# Freeze-frame imaging of synaptic activity using SynTagMA

Alberto Perez-Alvarez [1,8], Brenna C. Fearey [1,8], Ryan J. O'Toole [2], Wei Yang [3], Ignacio Arganda-Carreras[4,5,6], Paul J. Lamothe-Molina [1], Benjamien Moeyaert [7], Manuel A. Mohr[7], Lauren C. Panzera [2], Christian Schulze[1], Eric R. Schreiter [7], J. Simon Wiegert [3], Christine E. Gee [1], Michael B. Hoppa[2] & Thomas G. Oertner [1✉]

Information within the brain travels from neuron to neuron across billions of synapses. At any given moment, only a small subset of neurons and synapses are active, but finding the active synapses in brain tissue has been a technical challenge. Here we introduce SynTagMA to tag active synapses in a user-defined time window. Upon 395–405 nm illumination, this genetically encoded marker of activity converts from green to red fluorescence if, and only if, it is bound to calcium. Targeted to presynaptic terminals, preSynTagMA allows discrimination between active and silent axons. Targeted to excitatory postsynapses, postSynTagMA creates a snapshot of synapses active just before photoconversion. To analyze large datasets, we show how to identify and track the fluorescence of thousands of individual synapses in an automated fashion. Together, these tools provide an efficient method for repeatedly mapping active neurons and synapses in cell culture, slice preparations, and in vivo during behavior.

[1] Institute for Synaptic Physiology, University Medical Center Hamburg-Eppendorf, Hamburg D-20251, Germany. [2] Department of Biological Sciences, Dartmouth College, Hanover, NH 03755, USA. [3] Research Group Synaptic Wiring and Information Processing, University Medical Center Hamburg-Eppendorf, Hamburg D-20251, Germany. [4] Ikerbasque, Basque Foundation for Science, Bilbao, Spain. [5] Dept. of Computer Science and Artificial Intelligence, Basque Country University, San Sebastian, Spain. [6] Donostia International Physics Center (DIPC), San Sebastian, Spain. [7] HHMI, Janelia Farm Research Campus, Ashburn, VA 20147, USA. [8] These authors contributed equally: Alberto Perez-Alvarez, Brenna C. Fearey. ✉email: thomas.oertner@zmnh.uni-hamburg.de

The physical changes underlying learning and memory likely involve alterations in the strength and/or number of synaptic connections. On the network level, neuroscience faces an extreme 'needle in the haystack' problem: it is thought to be impossible, in practice, to create a map of all synapses that are active during a specific sensory input or behavior. Excellent genetically-encoded sensors for calcium, glutamate and voltage have been developed[1–3], which when combined with two-photon laser-scanning microscopy can monitor the activity of neurons even down to the synaptic level in highly light-scattering brain tissue[4]. However, the tradeoff between spatial and temporal resolution makes it impossible with this technology to simultaneously measure fluorescence in the thousands of synapses of even a single pyramidal neuron. Most functional imaging experiments are therefore limited to cell bodies, i.e., low spatial resolution[5], or monitor the activity of a few synapses within a single focal plane[6]. Multi-beam scanning designs have been proposed, but due to scattering of emitted photons, they do not produce sharp images at depth[7,8]. Projection microscopy can be a very efficient approach[9], but only in situations where the fluorescent label is restricted to one or very few neurons. In general, the need to choose between high temporal or high spatial resolution limits the information we can extract from the brain with optical methods.

A strategy to overcome this limit is to rapidly 'freeze' activity in a defined time window and read it out at high resolution later. The Ca$^{2+}$-modulated photoactivatable ratiometric integrator CaMPARI undergoes an irreversible chromophore change from green to red when the Ca$^{2+}$ bound form is irradiated with violet (390–405 nm) light[10,11]. CaMPARI has been successfully applied to map the activity of thousands of neurons in zebrafish, *Drosophila*, and in mouse[10,12–14]. As CaMPARI was designed to diffuse freely within the cytoplasm, it does not preserve subcellular details of Ca$^{2+}$ signaling. By anchoring CaMPARI to either pre- or postsynaptic compartments, we are able to mark active synapses a in short time window defined by violet light illumination. Three steps were necessary to create a Synaptic Tag for Mapping Activity (SynTagMA): (1) We introduced point mutations in CaMPARI to generate a new probe, CaMPARI2, with improved brightness and conversion efficiency[11]; (2) We target CaMPARI2 (F391W_L398V) to either presynaptic boutons by fusing it to synaptophysin (preSynTagMA) or to the postsynaptic density by fusing it to an intrabody against PSD95[15] (postSynTagMA); (3) We present an analysis workflow that corrects for chromatic aberration, tissue displacement (warping) and automatically finds regions of interest (i.e., postsynapses or boutons) to quantify green and red fluorescence. Bouton-localized preSynTagMA allows us to distinguish active and inactive axons. We use spine-localized postSynTagMA to visualize the extent of action potential back-propagation into the large apical dendritic tree of hippocampal pyramidal cells. Following repeated sparse activation of Schaffer collateral axons, postSynTagMA marks a small subset of synapses on CA1 pyramidal cells as highly active. PostSynTagMA is not only useful to analyze the organization of inputs on spiny pyramidal cells, but works equally well for interneurons where most synapses are formed on the dendritic shaft. A fraction of postSynTagMA is sequestered to the nucleoplasm, generating a useful label to identify neurons that are active during a specific behavior. As an application example, we photoconvert active CA1 pyramidal cells during reward collection in a spatial memory task. In summary, the key advantage of SynTagMA compared to acute calcium sensors is the greatly extended read-out period, allowing three-dimensional scanning of relatively large tissue volumes at cellular or at synaptic resolution.

## Results

**Creating and characterizing a presynaptic marker of activity.** To visualize activated presynaptic boutons, we fused the Ca$^{2+}$-modulated photoactivatable ratiometric integrator CaMPARI[10] to the vesicular protein synaptophysin (sypCaMPARI). In cultured hippocampal neurons, sypCaMPARI showed a punctate expression pattern along the axon, indicating successful targeting to vesicle clusters in presynaptic boutons (Fig. 1a). SypCaMPARI fluorescence decreased after stimulating neurons to evoke trains of action potentials (APs), saturating at 50 APs and slowly returning to baseline (Fig. 1b–d). This Ca$^{2+}$-dependent dimming, which is a known property of CaMPARI[10], provides a low-pass-filtered read-out of ongoing neuronal activity. Indeed, the signal-to-noise ratio was sufficient for detecting single APs in single boutons when 30 sweeps were averaged, suggesting a high sensitivity of localized sypCaMPARI for Ca$^{2+}$ influx (Fig. 1c). The Ca$^{2+}$-bound form of CaMPARI is irreversibly photoconverted from a green to a red fluorescent state by violet light irradiation[10]. Repeatedly pairing electrical field stimulation with 405 nm light flashes increased the red-to-green fluorescence ratio (R/G) of sypCaMPARI boutons in a stepwise fashion (Fig. 1e, f). Blocking action potential generation with tetrodotoxin (TTX) strongly reduced photoconversion, indicating that spike-induced Ca$^{2+}$ influx through high-threshold voltage-activated Ca$^{2+}$ channels was necessary for efficient conversion. Similar to the dimming response, the amount of sypCaMPARI photoconversion depended on the number of APs (Fig. 1g), suggesting that the R/G ratio can be interpreted as a lasting and quantitative record of axonal activity rather than just a binary signal.

Ca$^{2+}$-dependent dimming of sypCaMPARI peaked about 1 s after stimulation and returned to baseline after 3–15 s (Fig. 1b, c). To determine the photoconversion time window, multiple delays of violet light illumination were tested in single experiments using a digital mirror device (Fig. 2a). Photoconversion of sypCaMPARI was not efficient when light pulses (100 ms duration) were coincident with stimulation onset (Fig. 2b, c). Rather, maximal photoconversion occurred when violet light was applied 2–5 s after stimulation. The photoconversion window of sypCaMPARI extended to at least 10 s after stimulation, outlasting the dimming response to 5 APs (Fig. 1b). Some photoconversion also occurred in the absence of stimulation or in the presence of TTX (Fig. 1e, g and Fig. 2b, c). The long temporal window and activity-independent photoconversion are both undesirable traits that limit the utility of sypCaMPARI.

Parallel efforts to improve CaMPARI resulted in CaMPARI2, which contains a number of point mutations improving brightness, increasing kinetics, reducing activity-independent photoconversion and lowering the Ca$^{2+}$ affinity[11]. We selected the variant CaMPARI2 (F391W_L398V) ($K_d$ Ca$^{2+}$ = 174 nM, $k_{on}$ 66 s$^{-1}$, $k_{off}$ 0.37 s$^{-1}$, photoconversion rate in slices 0.22 s$^{-1}$ with Ca$^{2+}$, 0.0021 s$^{-1}$ without Ca$^{2+}$)[11] and fused it to synaptophysin to create preSynTagMA and expressed it in cultured hippocampal neurons. The temporal precision and dynamic range of preSynTagMA were both enhanced. PreSynTagMA showed no photoconversion in the absence of activity (Supplementary Fig. 1a–c). We could readily distinguish active from inactive axons by the differential preSynTagMA photoconversion in the presence of the GABA$_A$ antagonist bicuculline (Supplementary Fig. 1d–g). When the same axons were directly electrically stimulated via field electrodes, green preSynTagMA fluorescence dimmed in all axons, indicating that action potentials triggered calcium transients in all axons (Supplementary Fig. 1h). The rapid recovery from dimming corresponds to the short photoconversion time window (0.2–2 s post-stimulation) of preSynTagMA (Fig. 2d, e; Supplementary Fig. 1h). The photoconversion time windows of preSynTagMA and

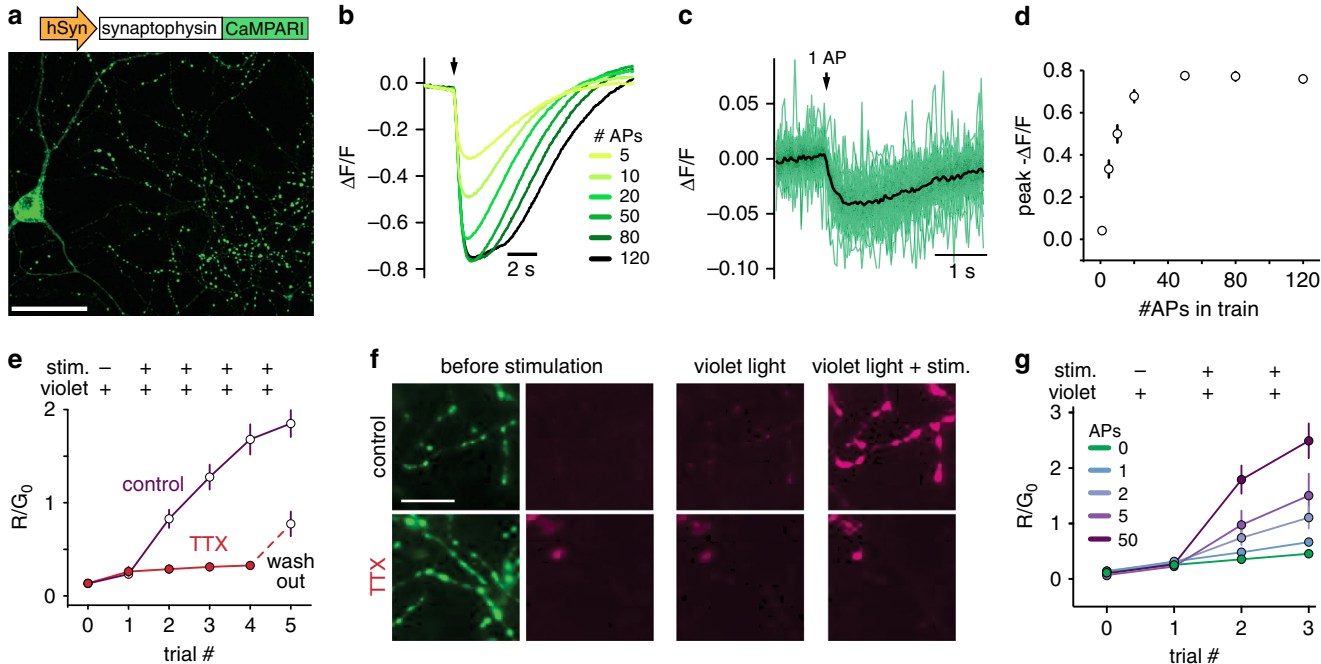

**Fig. 1 Synaptophysin-fused CaMPARI marks active presynaptic terminals. a** Representative image of cultured rat hippocampal neurons expressing sypCaMPARI. Note the clear punctate labeling of axonal boutons. **b** Average fluorescence response of sypCaMPARI boutons (green channel emission) to varying numbers of action potentials (APs) evoked at 50 Hz ($n = 6$ neurons, 317 synapses). **c** Trial-averaged responses to 30 single APs (green, $n = 57$ synapses). Black line is the average response of $n = 3$ neurons. **d** Plot of the maximum $\Delta F/F$ versus number of APs from the experiments in **b** and **c**. **e** Plot of initial red to green ratio of boutons expressing sypCaMPARI at baseline, after photoconverting violet light alone (20 light pulses of 1 s duration at 0.1 Hz, 405 nm, 10.8 mW cm$^{-2}$) and after simultaneous stimulation with trains of 50 APs at 50 Hz (trials 1–4). The experiment was performed in the absence or presence of 3 μM tetrodotoxin to block action potentials (control: $n = 8$ neurons; TTX: $n = 7$ neurons). Note that after washing out TTX, the R/G$_0$ ratio (trial 5) increased to the same amplitude as the first instance in control neurons (trial 2). **f** Representative red (magenta, trial 0, trial 1, trial 3) and green (green, trial 0) images of boutons from the experiment in **e**. **g** The amount of photoconversion (R/G$_0$) in a similar experiment as **e** but varying the number of APs in a 50 Hz train (20 light pulses of 1 s duration at 0.1 Hz, 405 nm, 54.1 mW cm$^{-2}$; Stim: 20 bursts at 50 Hz). AP fold increase was statistically different from all other stimulation conditions using a one-way ANOVA with Tukey's post-hoc comparison (*$p = 0.032$). Neurons per condition: 0 AP ($n = 4$), 1 AP ($n = 4$), 2 APs ($n = 5$), 5 APs ($n = 5$), 50 APs ($n = 7$). Data are presented as mean ± SEM in **d**, **e**, and **g**. Scale bar: 50 μm (**a**), 20 μm (**f**). Cultured neurons were 14–22 days old, expressing sypCaMPARI for 8–16 days.

CaMPARI2 (F391W_L398V) were near-identical (Fig. 2e), suggesting that synaptic targeting did not affect the kinetics of the indicator. Two hours after photoconversion, the R/G ratio was still 68% of the peak value, indicating photoconverted preSynTagMA marks activated boutons for several hours (Supplementary Fig. 1i). To quantify preSynTagMA localization, we co-expressed preSynTagMA together with cytosolic mCerulean in hippocampal slice cultures. In Schaffer collateral axons, preSynTagMA was 3.8-fold enriched in boutons vs. axonal shafts (Supplementary Fig. 2). For dense axonal labeling, we microinjected a viral vector (AAV2/9-syn-preSynTagMA) in CA3 and imaged Schaffer collateral boutons in CA1 *stratum radiatum*. Repeated electrical stimulation combined with violet light illumination induced photoconversion of preSynTagMA-expressing boutons while no photoconversion was induced by identical violet illumination when action potentials were suppressed by TTX (Supplementary Fig. 3).

**Targeting SynTagMA to excitatory postsynapses**. We chose to target SynTagMA to the postsynaptic protein PSD95, which has a higher retention time than most postsynaptic density proteins[16]. We decided against making a PSD95-CaMPARI fusion protein as overexpression of PSD95 is known to induce dramatic changes in function, size, and connectivity of dendritic spines[17]. Instead, we fused CaMPARI2 (F391W_L398V) (after deleting the nuclear export sequence) to a genetically encoded intrabody against PSD95 (PSD95.FingR)[15]. The resulting fusion protein, however, was not restricted to dendritic spines, where most excitatory

synapses are located, but labeled the entire dendrite of CA1 pyramidal neurons (Fig. 3a, d). We reasoned that the lack of spine enrichment was due to a large fraction of unbound cytoplasmic protein. An elegant method to reduce cytoplasmic fluorescence is to fuse a zinc finger (ZF) and the transcription repressor KRAB (A) to the targeted protein and include a ZF binding sequence near the promoter[15,18]. It is presumed that the ZF-KRAB domains direct excess unbound cytoplasmic protein into the nucleus where the ZF binds to the ZF binding sequence and KRAB(A) suppresses transcription of the exogenous genes. We added these additional regulatory elements to the PSD95.FingR-CaMPARI2 (F391W_L398V) construct, which has no additional nuclear export or localization sequences added, and co-expressed it with mCerulean. Punctate green fluorescence was now observed predominantly in spines and nuclei were fluorescent as expected for a ZF-KRAB(A) containing protein. The ratio of spine-to-dendrite green fluorescence was about 4 times higher than mCerulean spine-to-dendrite ratios, suggesting CaMPARI2 was now localized to postsynapses (Fig. 3b, e). Serendipitously, we discovered that the upstream ZF binding sequence was dispensable for autoregulation (Fig. 3c, f), which simplified swapping of promotors. We named this minimal construct postSynTagMA and characterized it further. To test for potential effects of postSynTagMA on neuronal physiology, we measured passive and active electrical properties, miniature excitatory postsynaptic currents and spine densities of postSynTagMA/mCerulean-expressing neurons and neurons expressing only mCerulean

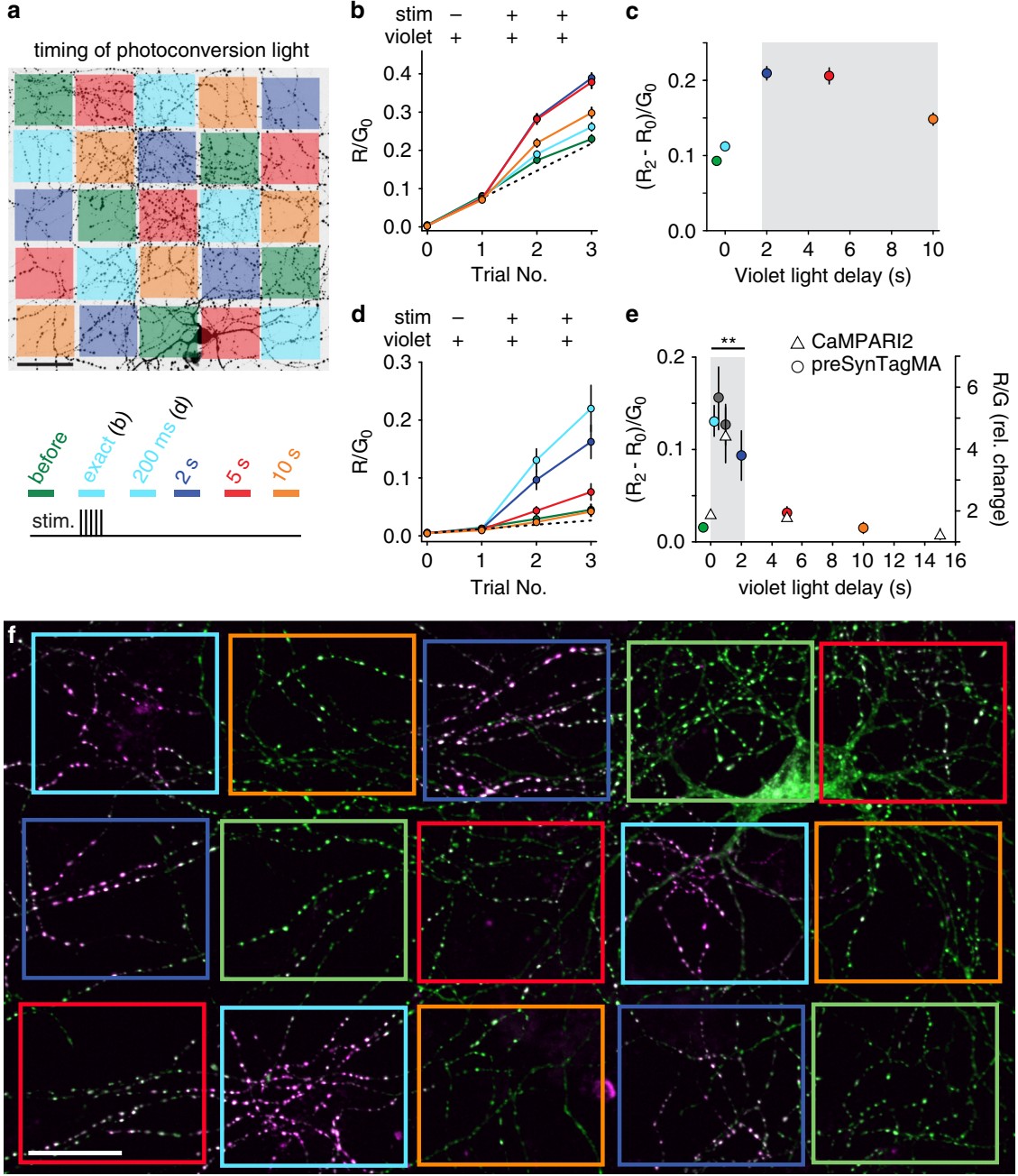

**Fig. 2 Temporal resolution of preSynTagMA photoconversion. a** A spatial light modulator was used to illuminate parts of the axonal arbor (405 nm, 50 mW cm$^{-2}$, 100 ms) at different times relative to a brief tetanic stimulation (5 APs). After each trial, new images were acquired. 'Exact' & '200 ms' timing share the same color code (cyan) as we used 'exact' timing in sypCaMPARI experiments and a 200 ms delay in preSynTagMA experiments. **b** Ratio of red to green fluorescence (R/G$_0$) from sypCaMPARI boutons illuminated at different times relative to the electrical stimulation. Trial 1 shows the effect of illumination without stimulation. Line color code as in **a**, $n = 6$ neurons. **c** Activity-dependent photoconversion ($\Delta$R/G$_0$) versus delay from start of stimulation to violet light from the same experiments in **b**. The gray box indicates the time window for efficient photoconversion of sypCaMPARI. **d** Neurons expressing preSynTagMA (synaptophysin-CaMPARI2 (F391W_ L398V)) were stimulated as in **a–c**. Note the greatly reduced increase in R/G$_0$ with violet light alone (Trial 1). $n = 6$ neurons. **e** Left axis: activity-dependent photoconversion ($\Delta$R/ G$_0$) versus delay of preSynTagMA expressing neurons (circles). $n = 12$ neurons for the before, 200 ms, and 2 s conditions and $n = 6$ neurons for the 500 ms, 1 s, 5 s and 10 s conditions. Right axis: photoconversion of CaMPARI2(F391W_L398V) after 50 bAPs and 100 ms violet light (triangles). $n = 5$ neurons. **f** Cultured rat hippocampal neuron expressing preSynTagMA, trial # 3 from **d**. Boxes indicate regions where photoconversion light was applied with different delays (color code as in **a**). All data are presented as mean ± SEM. In **e**, a Kruskal-Wallis test followed by the Benjamini-Hochberg FDR method was used. Timing conditions inside the gray box were not significantly different while the timing conditions outside the gray box were significantly different from those within the box ($p < 0.001$). Scale bars: 50 μm (**a**), 25 μm (**f**). Cultured neurons were 14–18 days old, expressing sypCaMPARI or preSynTagMA for 8–16 days.

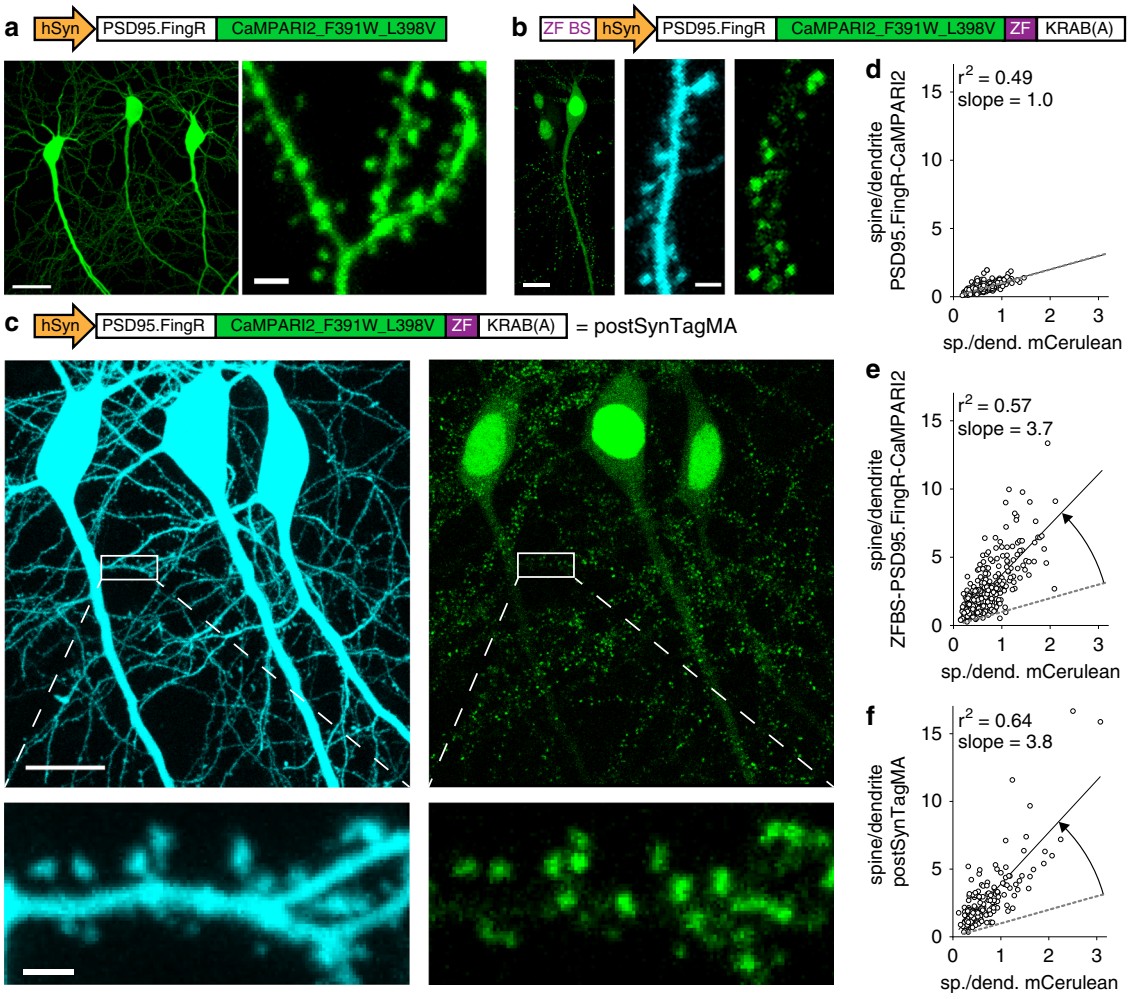

**Fig. 3 Postsynaptic targeting of SynTagMA using a PSD95 intrabody. a** CA1 neurons expressing the unregulated construct: a fusion protein of PSD95 fibronectin intrabody (PSD95.FingR) and CaMPARI2_F391W_L398V (without NES or epitope tags). Scale bars: 20 µm, 2 µm. **b** CA1 neurons expressing PSD95.FingR-CaMPARI2 with a zinc finger binding sequence (ZF BS) added upstream of the promoter and a zinc finger (ZF) fused to a transcriptional repressor domain (KRAB(A)) and mCerulean as a cytosolic filler (840 nm). Scale bars 12 µm (left panel) and 2 µm (center and right panels). **c** Postsynaptically targeted SynTagMA (postSynTagMA). As in **b**, with the ZF-KRAB(A) but no upstream ZF-BS. Note that postSynTagMA is still enriched in spines and the nucleus, leaving the cytoplasm almost free of SynTagMA. Scale bars: upper 20 µm, lower 2 µm. **d** The unregulated construct is expressed at high levels, leading to near-identical concentrations in dendrites and spines. For individual spines (circles), the spine-to-dendrite green fluorescence ratio was similar to the mCerulean spine-to-dendrite ratio ($n = 348$ spines, 4 neurons). **e** The construct with autoregulatory elements shuts off its own production, resulting in strong enrichment in spines ($n = 367$ spines, 3 neurons). **f** The construct without zinc finger binding sequence (postSynTagMA) is also auto-regulated, showing equally strong enrichment in spines ($n = 179$ spines, 3 neurons). All images are two-photon (2P) maximum intensity projections. In **d**, **e** and **f**, the black line is the linear fit to data points and the dotted line is the unity line.

(Supplementary Fig. 4). These control experiments, which were performed blind, yielded no significant differences between groups, indicating that expression of postSynTagMA did not alter neuronal physiology. For global labeling experiments, we generated a recombinant AAV2/9 encoding postSynTagMA, which produced dense punctate expression throughout the neuropil (Supplementary Fig. 5a–d). Viral expression of the construct without the autoregulatory elements (no ZF-KRAB(A)) flooded the neurons with fluorescent protein and was not usable for synaptic imaging (Supplementary Fig. 5e–g).

**Calibration of postSynTagMA with trains of action potentials.** We evoked backpropagating action potentials (bAPs) by brief somatic current injections to raise intracellular $Ca^{2+}$ and applied 395 nm light pulses (500 ms) with a 1 s delay to characterize postSynTagMA photoconversion. In the absence of stimulation (0 bAPs), violet illumination did not change the R/G

ratio ($R_1/G_1 = R_0/G_0$). Pairing violet light with increasing bAP trains (15 repeats) lead to increased $R_1/G_1$ ratios (Fig. 4a). To determine the best metric for quantifying SynTagMA photoconversion, we plotted several against the initial green fluorescence, which indicates synapse size (Fig. 4b, Supplementary Fig. 6). Considering only the change in red fluorescence ($\Delta R = R_1 - R_0$), the apparent photoconversion positively correlated with synapse size and only large synapses in the 3 bAP and 50 bAP groups could be separated from the non-converted (0 bAP) synapses. The $R_1/G_1$ ratio, albeit showing better separation than $\Delta R$, negatively correlated with size, and therefore would bias a classification of successful photoconversion towards small synapses. When strong photoconversion occurs, small synapses with few indicator molecules may lose all green fluorescence, which is problematic when one wishes to divide by this measure. Evidence of this is seen in the extremely high values (i.e., 20–200) for $R_1/G_1$ in small to medium-sized synapses (Fig. 4a). As

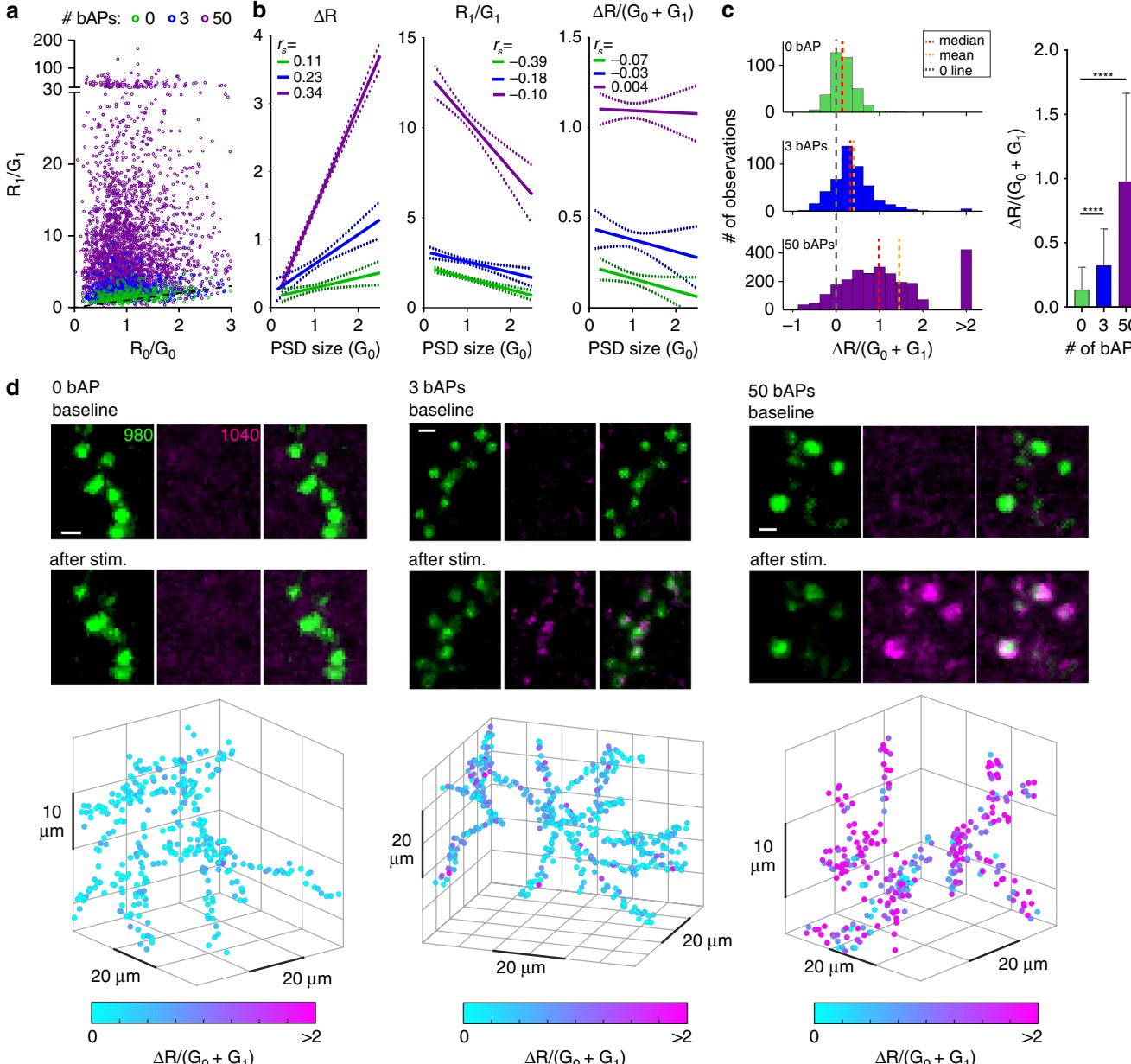

**Fig. 4 PostSynTagMA photoconversion with back-propagating action potentials (bAPs).** 2P image stacks of postSynTagMA expressing CA1 pyramidal neurons were taken before and after 15 pairings of trains of bAPs with photoconverting violet light (395 nm, 16 mW mm$^{-2}$, 500 ms duration, 1 s delay). Synaptic transmission was blocked. **a** R/G ratios of individual synapses before ($R_0/G_0$) vs after ($R_1/G_1$) photoconversion. Note the variability in photoconversion within conditions. Dotted black line is the unity line. Magenta: 50 bAPs. Blue: 3 bAPs. Green: 0 bAP. **b** Three different metrics vs. PSD size (linear regression lines with 95% confidence intervals, see Supplementary Fig. 6a-c for regressions). ΔR is positively correlated with PSD size (0 bAPs: $r_s = 0.11$, $p = 0.03$; 3 bAPs: $r_s = 0.23$, $p < 0.0001$; 50 bAPs: $r_s = 0.34$, $p < 0.0001$). Normalizing ΔR by ($G_0 + G_1$) removes the correlation with PSD size (0 bAP: $r_s = -0.07$, $p = 0.164$; 3 bAPs: $r_s = -0.03$, $p = 0.583$; 50 bAPs: $r_s = 0.004$ $p = 0.876$). $r_s$ is the correlation coefficient (Spearman's rho). **c** Histograms of photoconversion $\Delta R/(G_0 + G_1)$ for each condition (0 bAP: median = 0.139, mean = 0.164; 3 bAPs: median = 0.327, mean = 0.406; 50 bAPs: median = 0.984, mean = 1.452). Values above 2 are binned. Right bar graph: $\Delta R/(G_0 + G_1)$ vs the number of bAPs (data are median and interquartile range; Kruskal-Wallis test followed by Dunn's multiple comparsions: 0 bAP vs. 3 or 50 bAPs ****$p < 0.0001$, see Supplementary Fig. 6c for individual points). 0 bAP: $n = 356$ synapses, 1 neuron; 3 bAPs: $n = 472$ synapses, 1 neuron; 50 bAPs: $n = 2587$ synapses, 3 neurons. **d** Example green, red and merged images of SynTagMA-labeled synapses photoconverted with 0, 3 or 50 bAPs. 3D plots show the positions of individual synapses; markers are colored according to the calculated photoconversion value ($\Delta R/(G_0 + G_1)$). Scale bar: 1 μm.

photoconversion both increases red and decreases green fluorescence, we reasoned that using all channels before and after conversion would provide an optimal metric and avoid dividing by values close to 0. Indeed, $\Delta R/(G_0 + G_1)$ was not correlated with PSD size. Using this measure, we detected a significant

increase in photoconversion after $15 \times 50$ bAPs and even after $15 \times 3$ bAPs (Fig. 4c, d). The signal-to-noise ratio could be further improved by deconvolution of the image stacks (Supplementary Fig. 6d). To test whether violet light had adverse effects on cell health, we performed propidium iodide stainings of cultures

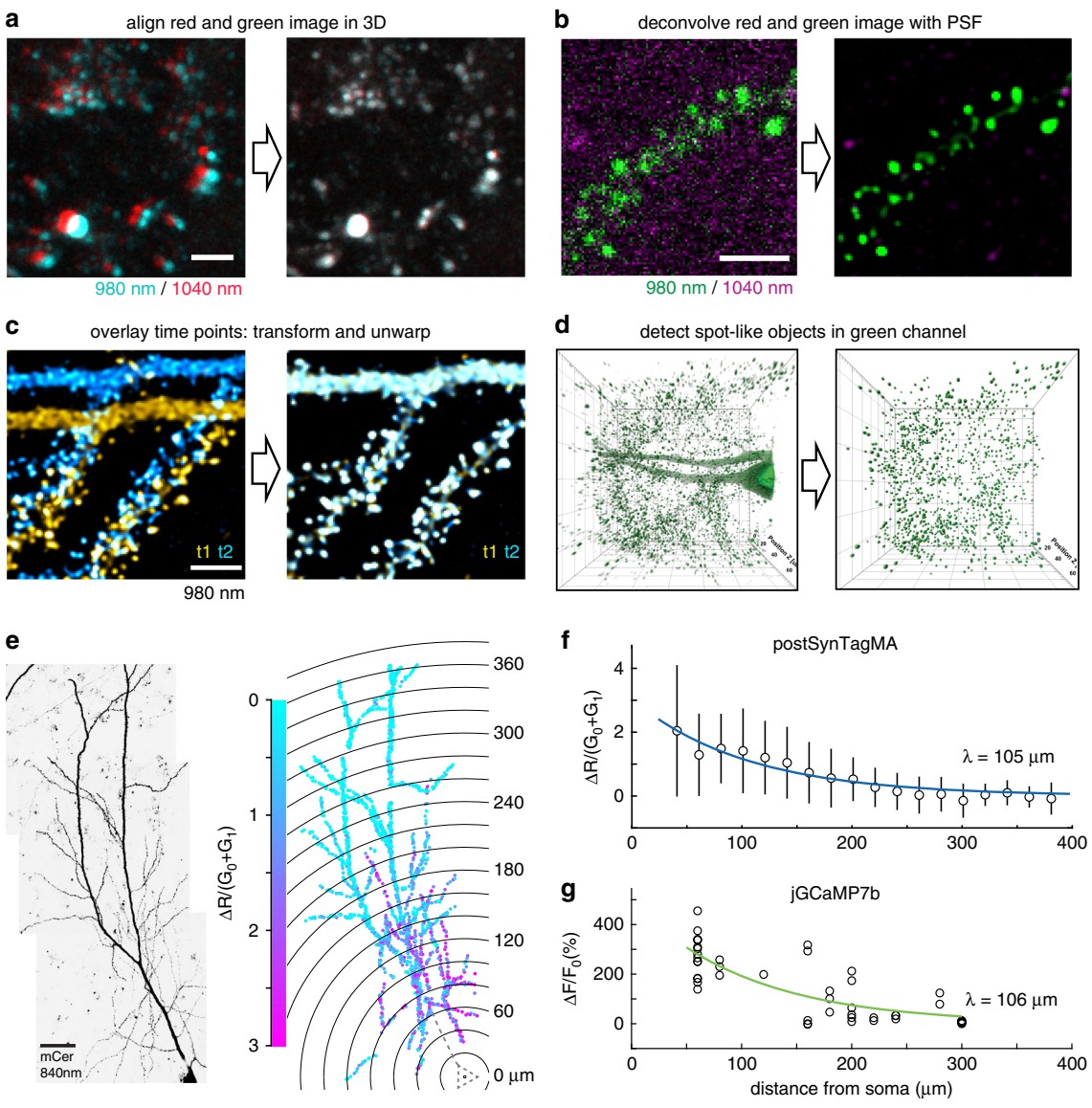

**Fig. 5 Workflow for automated detection and analysis of SynTagMA photoconversion. a** Green and red fluorescence is collected in alternate frames, switching between two Ti/Sapph lasers (980 nm/1040 nm). Images are registered in 3D to correct for chromatic aberration and laser alignment. **b** Median filtering and deconvolution is then applied to all images (both green and red channels). **c** To superimpose multiple time points in 3D, translation, rotation and unwarping are applied. **d** Synapses (ROIs) are detected as spherical objects, i.e., 'spots' from which fluorescence values are extracted and analyzed. **e** Maximum intensity projection of a CA1 pyramidal cell expressing cytosolic mCerulean (inverted gray scale) and postSynTagMA. The cell was stimulated with 50 bAPs at 100 Hz and illuminated with 395 nm (as in Fig. 4). For each identified synapse, $\Delta R/(G_0 + G_1)$ was analyzed, color-coded, and plotted at its original location. Distance from soma is indicated as concentric rings. **f** Photoconversion decreased exponentially from the soma with a distance constant of $\lambda = 105\,\mu m$ (median ± interquartile range, $n = 1860$ synapses, 1 cell, $R^2 = 0.91$). This experiment was reproduced in 4 neurons. **g** Spine calcium transient amplitudes during 50 bAP trains (jGCaMP7b) decreased exponentially with distance from the soma with $\lambda = 106\,\mu m$ ($n = 55$ synapses, 5 neurons, $R^2 = 0.66$). Scale bar: 4 μm (**a**, **b**, **c**), 30 μm (**e**).

exposed to different doses of violet light (Supplementary Fig. 7). The violet light dose used for SynTagMA photoconversion was well below the threshold for photodamage.

**Analysis workflow to quantify SynTagMA photoconversion.** While the efficient targeting of SynTagMA allows simultaneous interrogation of a large population of synapses, it also presents an analysis challenge. To place regions of interest (ROIs) on individual synapses (Figs. 1 and 2), simple spot detection algorithms (e.g., Imaris, ImageJ) can be used, but matching of objects across several time points is not straightforward. When the number of synapses reaches into the thousands, a manually curated approach is no longer feasible. We developed an image analysis workflow to tackle this problem. Images acquired using two-photon excitation

at different wavelengths must first undergo correction for laser alignment and chromatic aberration (Fig. 5a). As synapses in tissue are quite motile even on short time scales[19,20], a non-rigid 3D transformation, termed 'unwarping', was applied to re-align and preserve synapse identity across time points. To achieve the highest quality transformation, images were first deconvolved and then underwent 'unwarping' (Fig. 5b-c). Synapses were then detected on transformed datasets using the Spot feature of Imaris (Oxford Instruments) (Fig. 5d). Although each of these steps can be performed using a combination of freely and commercially available software packages, it is a time-consuming process that generates several gigabytes of data in intermediate steps. We therefore developed SynapseLocator (available at GitHub), a Matlab-based program to streamline the aforementioned steps

using freely available ancillary software tools (Fiji[21], Deconvolu-tionLab2[22], FeatureJ, elastix[23]). A machine-learning approach[24] was implemented to generate synapse templates (boutons or postsynaptic sites) that were used to automatically detect and extract fluorescence values. Transformed images and fluorescence values from identified synapses (ROIs) were saved for statistical analysis and imported into ImageJ or Imaris for 3D visualization (Fig. 5d). A specific difficulty for automated analysis is the pre-sence of spot-like autofluorescence with a broad emission spec-trum, leading to detection of spurious objects outside the labeled neurons. Objects with elevated $R_0$ values (10–30% of all detected objects) were considered non-synaptic and therefore excluded from further analysis (Supplementary Fig. 8). Automation of the synapse identification process greatly reduced the analysis time (from days for ~500 hand-curated ROIs at several time points to minutes for ~4000 automatically detected ROIs from the same dataset, using a personal computer). Manual and automated analysis produced comparable results (Supplementary Fig. 9).

**Back-propagating action potentials reach proximal synapses**. For a given number of bAPs, the amount of photoconversion was quite variable between individual spines (Fig. 4a), likely reflecting spine $Ca^{2+}$ transients of different amplitude. Whereas synapses close to the soma were strongly photoconverted, conversion decreased with distance, suggesting that bAP bursts do not increase $[Ca^{2+}]$ in distal branches of the apical dendrite (Fig. 5e, f). To test whether SynTagMA conversion is a linear reporter of local $Ca^{2+}$ signal amplitude, we performed equivalent experi-ments on CA1 neurons expressing jGCaMP7b. Indeed, $Ca^{2+}$ imaging during trains of bAPs yielded the same exponential decay of amplitude as a function of distance from the soma, validating our observation that bursts of bAPs do not invade distal dendrites (Fig. 5g). In contrast to imaging $Ca^{2+}$ synapse-by-synapse, postSynTagMA photoconversion reliably reports the amplitude of bAP-induced $Ca^{2+}$ elevations across the entire dendritic tree, resulting in higher coefficients of determination ($R^2$) in the sta-tistical analysis ($R^2 = 0.91$ vs. 0.66).

**Using postSynTagMA to map synaptic activity**. Having estab-lished the linear $Ca^{2+}$-dependence of photoconversion in patch-clamped CA1 neurons, we wanted to assess how long converted postSynTagMA would persist in individual spines. We illumi-nated rectangular areas of cultured neurons while stimulating the culture at 50 Hz, limiting postSynTagMA photoconversion to the illuminated dendritic sections (Fig. 6a). We observed that $\Delta R/(G_0 + G_1)$ returned to baseline with a time constant of 29 min (Fig. 6b), in contrast to the much slower turnover of pre-SynTagMA (Supplementary Fig. 1i) and of photoconverted soluble CaMPARI[11]. This relatively fast decay of the spine signal, which is consistent with the short retention time of synaptic PSD95 in the neocortex of young mice[25], limits the post-photoconversion acquisition time to about 30 min.

We next tested whether presynaptic stimulation would raise spine $Ca^{2+}$ sufficiently to trigger postSynTagMA photoconver-sion. In hippocampal slice culture, we activated Schaffer collateral axons and illuminated the dendrites of postSynTagMA-expressing CA1 pyramidal cells (100 ms violet light after 1 s delay, 50 repeats). Strong synaptic stimulation resulted in widespread photoconversion in dendritic spines (Fig. 6c). How-ever, as postsynaptic neurons may have been driven to spike in these experiments, we could not distinguish spines that received direct synaptic input from spines that were passively flooded with $Ca^{2+}$. Weak stimulation resulted in a very sparse and distributed conversion pattern, suggesting active synapses (Fig. 6d). In the example, a $65 \times 65 \times 78\,\mu m^3$ volume contained

1500 postSynTagMA-labeled synapses, fourteen of which showed values above 3σ ($\Delta R/(G_0 + G_1) = 0.97$) and were therefore classified as active. Given the relatively low release probability of Schaffer collateral synapses, only a third of this 1% of activated synapses will release transmitter at any given stimulation pulse. This low number of synchronized inputs is not expected to generate APs or dendritic spikes in the postsynaptic neuron, which is consistent with the distributed pattern of photocon-verted synapses.

To demonstrate that postSynTagMA indeed labels active synapses, it would be desirable to have an independent marker of synaptic activation. We turned to optogenetic stimulation to allow visualization of active presynaptic terminals. CA3 neurons expressing channelrhodopsin2 ET/TC (ChR2) and synaptophysin-mCerulean[26] were stimulated by blue light pulses (Fig. 6e). Light stimulation intensity was adjusted to be below the threshold for AP generation in a patch-clamped CA1 pyramidal cell ('reporter' neuron). Combining light stimulation of CA3 neurons with violet light illumination of postSynTagMA-expressing CA1 pyramidal cells resulted in very sparse photoconversion of Schaffer collateral synapses (Fig. 6f, g). Next to strongly photoconverted spines, cyan fluorescent boutons were observed, suggesting that these synapses were directly innervated and activated by ChR2-expressing presynaptic CA3 neurons (Fig. 6h). Non-photoconverted spines were also distant from activated terminals and thus represented 'true negatives' (Fig. 6i, Supplementary Movie 1). As red postSynTagMA has a relatively rapid turnover (Fig. 6b), it should be possible to generate multiple maps of active synapses over time. Indeed, using electrical stimulation of Schaffer collateral axons, we were able to convert (median $\Delta R/(G_0 + G_1) = 1.26$) and, 18 hours later, reconvert (median $\Delta R/(G_2 + G_3) = 1.25$) postSynTagMA expressing CA1 pyramidal cells (Fig. 6j). The remarkably similar degree of conversion on consecutive days suggests the possibility of repeated activity mapping with postSynTagMA.

**PostSynTagMA maps active neurons and synapses in vivo**. The nuclear sequestration of postSynTagMA prompted us to test whether it could be used to identify active neurons in vivo. After injection with AAV2/9-hSyn-postSynTagMA, and a chronic hippocampal imaging window was implanted, head-fixed mice were trained to run on a linear treadmill. While running, the hippocampus was illuminated with 405 nm light through the window; resulting in a small percentage of photoconverted CA1 neuronal nuclei (Fig. 7a). In the same mouse, there was no photoconversion during ketamine-xylazine anesthesia (Fig. 7b), consistent with the strongly reduced activity observed using GCaMP6f (Supplementary Movie 2). Next, we trained mice to stop and receive a water reward at a particular location (Fig. 7c, d). After reaching criterion, we imaged green fluorescence con-tinuously during four laps (Fig. 7e, f), followed by 15 laps with reward-triggered 405 nm light illumination (sans imaging). The neurons that were photoconverted also showed dimming (i.e., increased $[Ca^{2+}]$) just prior to each reward (Fig. 7f, magenta) whereas a matched number of randomly selected non-converted neurons did not (Fig. 7f, green). Therefore, in awake behaving animals, postSynTagMA photoconversion selectively labels behaviorally relevant neurons with high calcium transients (Supplementary Movie 3). In contrast to analysis of cytoplasmic, nuclear-excluded calcium indicators (e.g., GCaMP), which require analysis packages such as Suite2P[27], we found automatic segmentation and analysis of the nuclear SynTagMA trivial.

The density of neuropil labeling in these mice (Supplementary Fig. 5a) prompted us to switch to expressing postSynTagMA in interneurons (Supplementary Fig. 10) in order to test whether postSynTagMA could be used to identify individual active

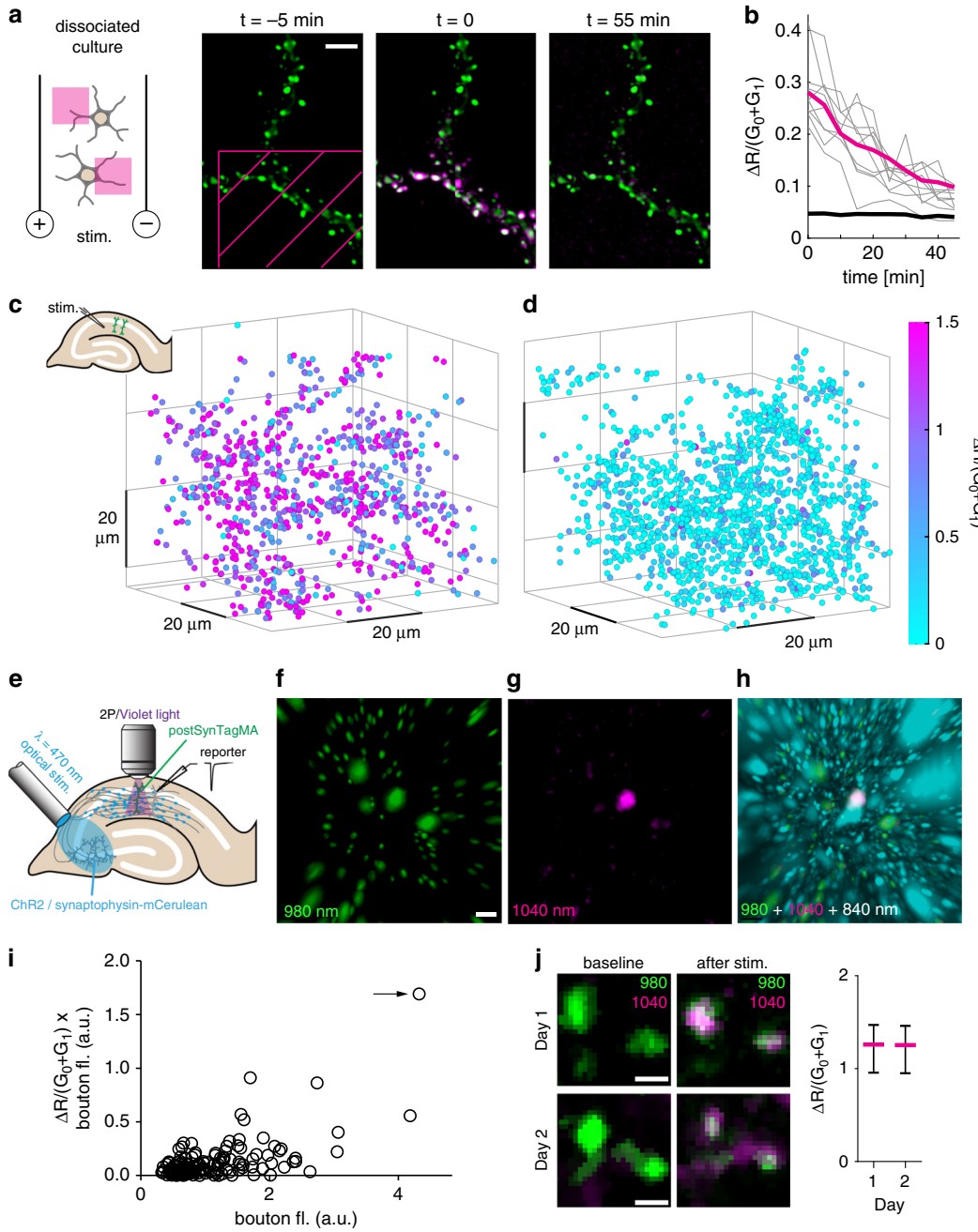

**Fig. 6 Using postSynTagMA to map active synapses. a** Time series from a postSynTagMA-expressing neuron in culture. At $t = 0$, one violet light pulse was applied via DMD inside the magenta hatched area (405 nm, 18.6 mW mm$^{-2}$, 0.5 s) while stimulating with a single train of action potentials delivered at 50 Hz for one second. **b** Inside the illuminated area, red fluorescence generated by photoconversion of postSynTagMA decayed exponentially with $\tau = 29.4$ min ($R^2 = 0.99$, $n = 138$ synapses, 2 experiments). Gray lines show individual synapses. For objects outside the illuminated square, $\Delta R/(G_0 + G_1)$ was constant (black line, $n = 712$ synapses, 2 experiments). **c** Photoconversion after Schaffer collateral stimulation in organotypic slice culture. Color-coded $\Delta R/(G_0 + G_1)$ of synapses plotted at their locations in the $65 \times 65 \times 78\,\mu$m volume of tissue. Strong stimulation of synaptic inputs was paired with violet light (100 ms, 395 nm, 16 mW mm$^{-2}$, 50 repeats). Note that $\Delta R/(G_0 + G_1)$ is equal or greater than 1.5 for most of the synapses ($n = 897$ synapses). **d** As in **c**, but with weak stimulation, below the threshold for postsynaptic APs ($n = 1502$ synapses). **e** CA3 neurons expressing ChR2-ET/TC and synaptophysin-mCerulean were stimulated with 470 nm flashes to evoke subthreshold (~500 pA) excitatory synaptic responses in a patch-clamped CA1 neuron (reporter). Simultaneously, an adjacent postSyntagMA-expressing CA1 neuron was illuminated with violet light (same illumination as **c-d**). **f** Green spots ($G_0 + G_1$) segmented with Imaris™. **g** Red fluorescence after photoconversion (R1), masked by the green channel. Note the strongly photoconverted synapse (spot #1). **h** Spot #1 (magenta) is in close contact with a presynaptic ChR2-expressing terminal (cyan). Non-converted (green) spots are not in direct contact with cyan terminals. **i** Quantification of panels **f-h** ($n = 167$ spots). Arrow indicates spot #1. **j** Red/green overlay before and after photoconversion of the same synapses on successive days. Strong extracellular stimulation was paired with 395 nm light (500 ms, 1 s delay, 15×). Photoconversion of postSynTagMA was near-identical on day 1 and day 2 ($p = 0.386$, paired two-tailed $t$-test, $n = 37$ synapses, data presented as median ± interquartile range). All experiments were reproduced in at least twice independent experiments. Scale bars: 5 μm (**a**), 3 μm (**f-h**), 1 μm (**j**).

synapses/dendrites in sparsely labeled neurons in vivo. We photoconverted sparsely labeled interneurons in *stratum oriens* under isoflurane anesthesia, which retains neuronal activity close to awake levels but reduces motion artifacts (Fig. 7g, h, Supplementary Movie 2). Photoconversion of clusters of synapses was observed; consistent with local dendritic calcium events that have been described in interneurons in vivo[28]. Thus, under conditions of sparse SynTagMA expression, it is possible to resolve individual photoconverted synapses in vivo. During photoconversion, movement of the brain is not problematic as imaging is not concurrent and SynTagMA can be used to tag active neurons and synapses during behavior. To resolve SynTagMA at the level of individual synapses, minimizing tissue movement with anesthesia may improve data quality.

## Discussion

To create a synaptically localized, photoconvertible calcium sensor, we fused CaMPARI2 to the vesicle protein synaptophysin (preSynTagMA) for presynaptic targeting or to an intrabody against the postsynaptic scaffold of excitatory synapses (postSynTagMA). Both targeting strategies have previously been used to label synapses with fluorescent proteins[15,29], and we verified that SynTagMA-expressing neurons are physiologically normal. When SynTagMA is combined with a second fluorescent protein, here mCerulean, postSynTagMA puncta can be assigned to particular dendrites or neurons. PreSynTagMA should prove useful for identifying active axons in tissue and may also prove useful for distinguishing high from low release probability boutons based on the graded photoconversion. PostSynTagMA should likewise facilitate identification and mapping of active synapses, but on the postsynaptic side.

Identifying active synapses is a long-standing quest in neuroscience and many approaches have been put forward. In two-dimensional neuronal cultures, it is relatively straightforward to monitor the activity of a large number of synapses with a high speed camera, e.g. by employing genetically encoded sensors of vesicular pH[30,31]. In intact brain tissue, however, monitoring more than one synapse with high temporal resolution becomes very challenging[6,32]. To create a more permanent tag of active synapses, $Ca^{2+}$ precipitation in active spines was used to create an electron-dense label that can be detected by electron microscopy[33]. As this method requires tissue fixation, it does not allow monitoring changes in synaptic activity over time. Expressing the two halves of split-GFP on the extracellular surface of pre- and postsynaptic neurons (mGRASP) labels contact points between the neurons as green fluorescent puncta[34]. Refined transsynaptic labeling methods with multiple colors have led to remarkable insights about network connectivity[35,36]. Whereas mGRASP and eGRASP just report anatomical proximity, SynTagMA photoconversion is $Ca^{2+}$-dependent and therefore sensitive to synaptic activity.

One of the many interesting applications of postSynTagMA will be to create extensively detailed input maps of individual neurons at single synapse resolution. The question of whether inputs carrying similar information segregate to branches of the dendritic tree is currently being investigated using spine-by-spine $Ca^{2+}$ imaging[6,37,38]. By freezing the activity status of all spines during a particular labeling protocol, SynTagMA may simplify such experiments by reading out fluorescence ratios in thousands of spines in a single high-resolution stack. Importantly, postSynTagMA maps can be repeated, opening the possibility to study the functional dynamics of excitatory connections rather than just morphology and turnover of dendritic spines. In addition to synapses on mushroom-shaped spines, postSynTagMA provides access to synapses on structures that are difficult to probe with

$Ca^{2+}$ dyes or diffusible GECIs, such as stubby spines or shaft synapses[39]. Thus, postSynTagMA may open the possibility to study the long-term dynamics of interneuron excitation, a key homeostatic mechanism that involves $Ca^{2+}$-induced-$Ca^{2+}$ release from intracellular stores[40]. In addition to $Ca^{2+}$ influx though voltage- or ligand-gated channels, $Ca^{2+}$ release is thought to be important in both presynaptic[41] and postsynaptic compartments[42–44], and SynTagMA may help investigating the sources of $Ca^{2+}$ in individual synapses.

To test the linearity of calcium-dependent photoconversion, we used bursts of action potentials (APs) which are known to be strongly attenuated on their way into the dendrite[45,46]. Indeed, postSynTagMA photoconversion was greatest in spines relatively close to the soma while distal synapses were not converted at all. The distance constant estimated from a Sholl analysis of postSynTagMA conversion ($\lambda = 105$ μm, Fig. 5f) was the same as that determined by calcium imaging of individual spines ($\lambda = 106$ μm, Fig. 5g). Previous studies[45,46] have measured back-propagation along the apical dendrite, but not along thin oblique dendrites or into individual spines. Studies with single-spine resolution[47,48] have not included large-scale spatial information, as only few synapses per cell can be investigated. In future SynTagMA studies, reconstructions of the neuronal morphology in 3D would allow accurate measurements of the dendritic path length to every synapse and combine that data with functional information. We observed that even close to the soma, conversion of spines was not very uniform and the same was true for boutons. Understanding the sources of this heterogeneity, and whether these correspond to particular classes of synapses, may lead to important insights into synapse-specific calcium regulation.

An unexpected application arose from the nuclear localization of postSynTagMA. As we show, nuclear SynTagMA conversion can be used to visualize the location of the most active neurons in behaving animals. As the cytoplasm is almost free of label, automatic segmentation is very easy. Nuclear calcium elevations may be of particular relevance during learning, as they are driven by burst firing and trigger transcription of plasticity-related genes[49]. There are several existing strategies for labeling active neurons[50], but their time resolution is in the hours timescale in contrast to the 2 s photoconversion time window of SynTagMA. SynTagMA might serve as an intermediate between acute calcium imaging and labeling methods based on immediate early gene expression.

There are some caveats to working with SynTagMA. Before or during imaging, it is important to photoswitch the protein to the bright state using low intensity violet light[10,11]. The quite rapid turnover of postSynTagMA means that after photoconversion, images should be collected within about 30 min. SynTagMA analysis is complicated by endogenous red fluorescence which can be mistaken for photoconverted SynTagMA. We found that depending on the age of the tissue, 0.4-4% of the total imaged volume was red fluorescent at baseline. Green objects detected inside these autofluorescent areas were typically non-synaptic (Supplementary Fig. 8). For automatic exclusion, SynapseLocator sets a dynamic threshold based on the histogram of the red channel at baseline ($R_0$, before photoconversion) and rejects objects with elevated $R_0$. This process effectively prevents false-positive errors, but requires acquisition of at least two time points (before and after photoconversion). If for experimental reasons only a single time point can be acquired, we strongly recommend excluding objects that have high red fluorescence in the voxels surrounding green puncta. When acquiring two-photon image stacks, we collected each optical section twice, exciting either the green or the red form of SynTagMA by rapidly switching the power of two different Ti:Sapph lasers. Near-simultaneous acquisition of red and green fluorescence prevented movement

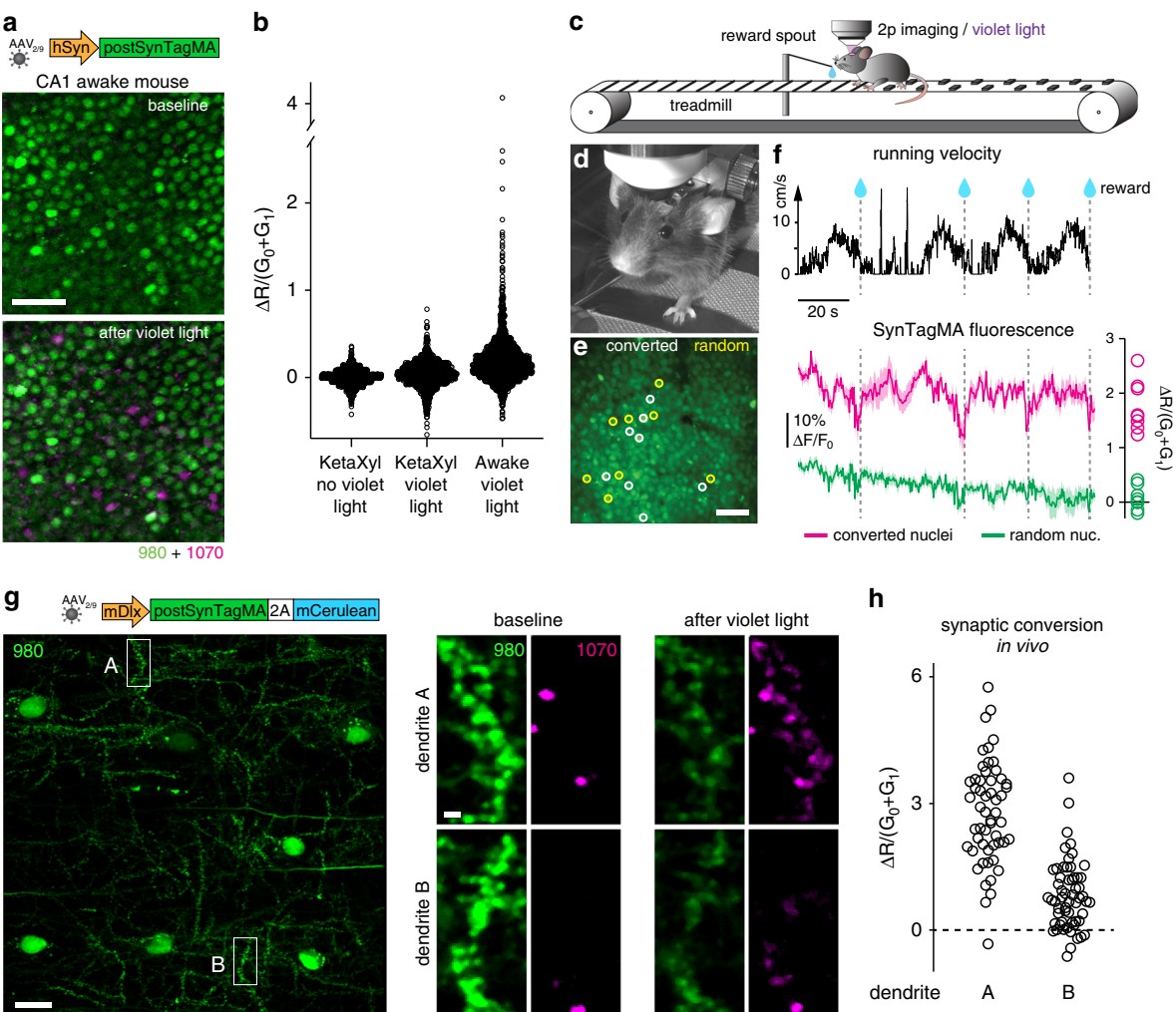

**Fig. 7 SynTagMA identifies active neurons during behavior. a** Nuclei of CA1 neurons expressing AAV2/9-hSyn-postSynTagMA imaged in vivo through a chronic cranial window using 980 nm and 1070 nm to excite green and red SynTagMA fluorescence, respectively. Ten 2 s, 405 nm, 12.1 mW mm$^{-2}$ light pulses were applied in an awake mouse after which a small percentage of nuclei became photoconverted (magenta). **b** Photoconversion relative to baseline under ketamine-xylazine anesthesia ($n = 1105$ nuclei), after violet light ($n = 1105$ nuclei) and under awake conditions ($n = 1576$ nuclei). **c** Closed loop paradigm: a head-fixed mouse was trained to stop at a certain position on the running belt to receive a water reward (teardrop). Nuclear fluorescence in CA1 was continuously monitored during 4 laps, followed by 15 laps where 405 nm, 12.1 mW mm$^{-2}$, 2 s light pulses were triggered during water reward. **d** Mouse engaged in the task. Note spout for water delivery. **e** 2P image of CA1 cell body layer during running. Eight nuclei that later became photoconverted are marked by white circles. Yellow circles are eight randomly selected non-converted nuclei used for comparison of calcium signals. **f** Black trace is running speed during the first 4 laps with times of reward delivery (teardrop/dashed line). Magenta and green traces are the average green SynTagMA fluorescence of the 8 photoconverted nuclei and the 8 non-converted nuclei indicated in **e**, respectively. At right is the photoconversion of the individual nuclei. Note the consistent dips in the magenta trace (i.e., high calcium) just before the water reward/photoconversion light would be triggered. **g** Interneurons expressing AAV2/9-mDlx-postSynTagMA-2A-mCerulean imaged through a chronic cranial window under isoflurane anesthesia (mCerulean fluorescence not shown). At high magnification, green fluorescence reveals PSD spots on the dendrite. Red spots at baseline are autofluorescent material, unrelated to SynTagMA. Violet light (20 flashes, 0.2 Hz, 3 s duration, 0.42 mW mm$^{-2}$) was applied to photoconvert postSynTagMA. **h** Photoconversion of synapses on dendrite A ($n = 54$ synapses) and B ($n = 58$ synapses) indicate higher activity levels in dendrite A. Scale bars: 50 μm (**a, e**); 20 μm and 2 μm (**g**). All experiments were performed in at least two mice and found to be reproducible.

artefacts and made it straightforward to correct for chromatic aberration. When the two color channels are collected in successive stacks, Ca$^{2+}$-dependent dimming driven by spontaneous activity may affect one and not the other channel. Particularly when imaging in vivo, the time required to re-tune a single laser to the different excitation wavelengths might render analysis of individual synapses impossible.

For in vivo experiments, depth-dependent attenuation of the violet photoconversion light by scattering and absorption has to be considered. We found that photoconversion efficiency was a linear function of violet light intensity (Supplementary Fig. 11). Using published modeling software[51], we compared cranial window

illumination (405 nm) to illumination via implanted optical fiber. While the relatively homogenous illumination through a cranial window leads to uniform light intensity (and thus, photoconversion) within a field of view (Fig. 7a, Supplementary Fig. 11), photoconversion via fiber would be expected to be rather inhomogeneous in all spatial directions and results therefore difficult to interpret. Similar to CaMPARI experiments, it is safest to compare cells or synapses imaged at similar depths[52].

If the postsynaptic neuron fires APs during strong (suprathreshold) synaptic activation, the resulting global calcium elevation could interfere with SynTagMA-based input mapping at proximal synapses. As we show (Fig. 5e), this is much less of a

problem in the distal dendritic tree. For input-mapping experiments, it is advisable to repeat weak, subthreshold stimuli many times, each time paired with a (delayed) pulse of violet light. To avoid phototoxicity, it is important to keep track of the total light dose applied during each protocol, using shorter pulses (or lower light intensities) for protocols with many repetitions.

Finally, SynTagMA is amenable for viral delivery using recombinant AAVs. Using different promoters, we demonstrate pan-neuronal expression for network analysis (Fig. 7a–f, Supplementary Fig. 3, Supplementary Fig. 5) and sparse viral expression of Syn-TagMA in hippocampal interneurons (Fig. 7g, Supplementary Fig. 10). To achieve sparse expression of SynTagMA in pyramidal cells, dual-AAV labeling systems could be employed[53].

## Methods

**Dissociated rat hippocampal cell cultures**. Neurons from P1 Sprague-Dawley rats of either sex were isolated from hippocampal CA1-CA3 regions with dentate gyrus removed, dissociated (bovine pancreas trypsin; 5 min at room temperature), and plated on polyornithine-coated coverslips inside a 6 mm diameter cloning cylinder. Calcium phosphate-mediated gene transfer was used to transfect 5-7 day old cultures. All measurements unless otherwise noted, are from mature 13–21 day old neurons. Procedures with Sprague-Dawley rats were approved by Dartmouth College's Institutional Animal Care and Use Committee (IACUC). Cells were maintained in culture media consisting of Earle's MEM (Thermofisher 51200038), 0.6% glucose, 0.1 g l$^{-1}$ bovine transferrin, 0.25 g l$^{-1}$ insulin, 0.3 g l$^{-1}$ glutamine, 5% fetal calf serum (Atlanta Biologicals), 2% B-27 (Life Technologies), and 4 μM cytosine β-D-arabinofuranoside added 48 h after plating in 6 mm diameter cloning cylinders (Ace Glass).

**Rat hippocampal slice cultures**. Hippocampal slice cultures from Wistar rats of either sex were prepared at day 4–7[54]. Briefly, rats were anesthetized with 80% $CO_2$ 20% $O_2$ and decapitated. Hippocampi were dissected in cold slice culture dissection medium containing (in mM): 248 sucrose, 26 NaHCO₃, 10 glucose, 4 KCl, 5 MgCl₂, 1 CaCl₂, 2 kynurenic acid, 0.001% phenol red (310–320 mOsm kg$^{-1}$, saturated with 95% O₂, 5% CO₂, pH 7.4). Tissue was cut into 400 μM thick sections on a tissue chopper and cultured at the medium/air interface on membranes (Millipore PICMORG50) at 37 °C in 5% CO₂. No antibiotics were added to the slice culture medium which was partially exchanged (60–70%) twice per week and contained (for 500 ml): 394 ml Minimal Essential Medium (Sigma M7278), 100 ml heat inactivated donor horse serum (H1138 Sigma), 1 mM L-glutamine (Gibco 25030-024), 0.01 mg ml$^{-1}$ insulin (Sigma I6634), 1.45 ml 5 M NaCl (S5150 Sigma)), 2 mM MgSO₄ (Fluka 63126), 1.44 mM CaCl₂ (Fluka 21114), 0.00125% ascorbic acid (Fluka 11140), 13 mM D-glucose (Fluka 49152). Wistar rats were housed and bred at the University Medical Center Hamburg-Eppendorf. All procedures were performed in compliance with German law and according to the guidelines of Directive 2010/63/EU. Protocols were approved by the Behörde für Gesundheit und Verbraucherschutz of the City of Hamburg.

**Mouse acute hippocampal slices**. Male, adult (4-6 months old) C57BL/6J mice were housed and bred in pathogen-free conditions at the University Medical Center Hamburg-Eppendorf. The light/dark cycle was 12/12 h and the humidity and temperature were kept constant (40% relative humidity; 22 °C). All procedures were performed in compliance with German law and according to the guidelines of Directive 2010/63/EU. Protocols were approved by the Behörde für Gesundheit und Verbraucherschutz of the City of Hamburg. A bilateral intrahippocampal injection with AAV2/9 encoding syn-postSyntagMA (3.4 × 1013 vg ml$^{-1}$) was administered 7–15 days before acute slice preparation. Mice were anesthetized by CO₂ inhalation, decapitated and brains rapidly removed. Coronal slices (300 um) were cut on a vibratome (Leica VT1000S) in ice cold solution containing (in mM): 110 choline chloride, 25 NaHCO₃, 25 D-Glucose, 11.6 sodium-L-ascorbate, 7 MgSO₄, 1.25 NaH₂PO₄, 2.5 KCl, and 0.5 CaCl₂. Slices were then incubated at 34 °C for 30–45 min in oxygenated acute slice artificial cerebrospinal fluid (ACSF), containing (in mM): 125 NaCl, 26.2 NaHCO₃, 11 D-Glucose, 1 NaH₂PO₄, 2.5 KCl, 1.3 MgCl₂, and 2.5 CaCl₂. Slices were kept in the same solution at room temperature until use.

**Anesthetized and awake mice**. Male, adult (4–6 months old) C57BL/6J mice were housed and bred in pathogen-free conditions at the University Medical Center Hamburg-Eppendorf. The light/dark cycle was 12/12 h and the humidity and temperature were kept constant (40% relative humidity; 22 °C). An AAV2/9 encoding mDlx-postSynTagMA (2.0 × 10¹² vg ml$^{-1}$), syn-postSynTagMA (3.4 × 10¹³ vg ml$^{-1}$) or syn-GCaMP6f (1.45 × 10¹³ vg ml$^{-1}$) was injected unilaterally in the hippocampus. All procedures performed in mice were in compliance with German law and according to the guidelines of Directive 2010/63/EU. Protocols were approved by the Behörde für Gesundheit und Verbraucherschutz of the City of Hamburg.

**Plasmid construction**. To create preSynTagMA, synaptophysin-GCaMP3 was, as previously described[55], digested by HindIII and BamHI to remove GCaMP3, and CaMPARI was fused to synaptophysin using the In-Fusion® HD Cloning method and kit (Takara Bio USA) after amplifying variants of CaMPARI with PCR using custom primers with base pair overhangs homologous to the synaptophysin plasmid (3' Primer [CGATAAGCTTTTATGAGCTCAGCCGACC], 5' Primer [CAGATGAAGCTTATGCTGCAGAACGAGCTTG]).

Developing postSynTagMA involved the creation of several intermediate constructs. After removal of restriction site XbaI from pCAG_PSD95.FingR-eGFP-CCR5TC, PSD95.FingR-eGFP-CCR5TC was inserted into a pAAV-hsyn1 backbone without the CAG promoter or upstream zinc finger binding site, to produce pAAV-syn-PSD95.FingR-eGFP-CCR5TC (available upon request). The eGFP was then replaced with CaMPARI[10], from which we had deleted the nuclear export signal (NES), to produce pAAV-syn-PSD95.FingR-dNES-CaMPARI1-CCR5TC, a fusion construct of the fibronectin intrabody and a CaMPARI variant that is not restricted to the cytosol and can enter the nucleus. To finally generate pAAV-syn-postSynTagMA, CaMPARI1 was then replaced by CaMPARI2 (without NES and epitope tags) and the point mutations F391W and L398V were introduced using QuickChange PCR to increase calcium affinity[11]. The left-handed zinc finger (aka CCR5TC) fused to the KRAB(A) transcriptional repressor[18] was removed to produce pAAV-syn-PSD95.FingR-dNES-CaMPARI2_F391W_L398V, the unregulated variant. A sequence including the zinc finger binding sequence[15,56] (5'-GTCATCCTCATC-3') upstream of the hsyn1 promoter was synthesized (ThermoFisher) and inserted using the MluI and EcoRI restriction sites to generate pAAV-ZFBS-syn-PSD95.FingR-dNES-CaMPARI2_F391W_L398V-CCR5TC. PostSynTagMA (ID: 119736) and preSynTagMA (ID: 119738) are available on www.addgene.org. Other plasmids such as the lower affinity pAAV-syn-PSD95. FingR-dNES-CaMPARI2-CCR5TC, pAAV-syn-synaptophysin-CaMPARI2 and the variants with zinc finger binding sequence are available upon request.

mDlx-postSynTagMA-2A-mCerulean was created by first removing the stop codon from postSynTagMA and inserting a 3'-BsiWI restriction site via PCR. 2A-mCerulean was then inserted 3'of postSynTagMA via Acc65 (5') and HindIII (3') to create postSynTagMA-2A-mCerulean. This construct was subsequently cloned via NheI (5') and BsrGI (3') into the mDlx-GFP-Fishell1 backbone by replacing the GFP with postSynTagMA-2A-mCerulean to generate mDlx-postSynTagMA-2A-mCerulean.

**Cell culture imaging**. SypCaMPARI experiments (Fig. 1) were performed at 34 °C using a custom-built objective heater. Coverslips were mounted in a rapid-switching, laminar-flow perfusion and stimulation chamber on the stage of a custom-built laser microscope. The volume of the chamber was maintained at ~75 μl and was perfused at a rate of 400 μl min$^{-1}$. Neurons were perfused continuously during imaging with a standard saline solution containing the following in mM: 119 NaCl, 2.5 KCl, 2 CaCl₂, 2 MgCl₂, 25 HEPES, 30 D-Glucose, 10 μM CNQX, and 50 μM D-APV. When noted, 3 μM Tetrodotoxin (TTX) and 20 μM Bicuculline were added to the saline solution.

Neurons were imaged through an EC Plan-Neofluar 40×1.3 NA objective (Zeiss) or an UAPON40XO340-2 40×1.35 NA objective (Olympus), using an IXON Ultra 897 EMCCD camera (Andor) at a frame rate of 25 Hz (exposure time: 39.72 ms). Green fluorescence was excited at 488 nm (Coherent OBIS laser, ~ 3 mW) through a ZET488/10x filter and ZT488rdc dichroic (Chroma). Red fluorescence was excited at 561 nm (Coherent OBIS) through a ZET561/10x filter and ZT561rdc dichroic (Chroma). Green and red fluorescence was collected via ET 525/50 m and ET600/50 m emission filters (Chroma), respectively.

**Cell culture stimulation and photoconversion**. Field stimulation-evoked action potentials were generated by passing 1 ms current pulses, yielding fields of ~12 V cm$^{-2}$ through the recording chamber bracketed by platinum/iridium electrodes. Electrical stimuli were locked to start according to defined frame number intervals using a custom-built board named "Neurosync" powered by an Arduino Duo chip (Arduino) manufactured by an engineering firm (Sensostar)[57]. A collimated 405 nm LED light source (Thorlabs) was set on top of the microscope stage with a custom-built plastic case (Bob Robertson, Dartmouth College). This light source was coupled to a T-Cube LED Driver (Thorlabs) and a Pulse Pal (Open Ephys) was used to trigger light flashes of specific duration and delay. The light source trigger was set relative to a TTL input from Neurosync. Power density of the 405 nm light was measured using a digital handheld optical power and energy meter with an attached photodiode power sensor (Thorlabs). The power densities used were either 10.8 mW cm$^{-2}$ or 54.1 mW cm$^{-2}$.

For incubator photoconversion experiments (Supplementary Fig. 1), custom-built circular 51-diode 405 nm LED arrays (Ultrafire) were wired up to a custom dual programmable relay board to flash light for 100 ms every 10 s, inside a cell culture incubator (New Brunswick; Eppendorf) set to ~37° C and ~5% CO₂. Neurons were then mounted on the microscope with perfusion, and green fluorescence was imaged during 50 stimulations @ 50 Hz (as above) to measure calcium-dependent dimming. Only neurons responsive to stimulation were then analyzed (over 90% of cells measured).

**Experimental setup for variably timed photoconversion**. Variably timed photoconversion experiments (Fig. 2) were performed on a Nikon Ti-E microscope

fitted with an Andor W1 Dual Camera (Andor CMOS ZYLA), dual spinning disk, Coherent Lasers (OBIS 405, 488 and 561 nm) and the Andor Mosaic 3 micromirror system, controlled by Andor iQ software, and Nikon elements for image acquisition. PulsePal software was used to time lock the stimulus with the mosaic sequence start. A custom 5×5, 250 $\mu m^2$ square grid was drawn to illuminate different areas (squares) at different times relative to the electrical stimulation. Photoconversion was induced using a train of 5 action potentials (50 Hz) paired with a 100 ms violet light pulse (405 nm, 50 mW cm$^{-2}$) at different delays. The protocol was repeated 5 times at 30 s intervals.

**Cell culture image analysis**. EMCCD camera images or confocal image stacks were imported into Fiji. A maximum intensity projection was made from the confocal stacks. Ten-pixel diameter circular ROIs were placed over boutons identified by eye using an ImageJ plugin (https://imagej.nih.gov/ij/plugins/time-series.html) to localize them over the brightest pixel in the green channel. Boutons were identified as punctate spots that showed a dimming response to AP stimulation (>98% of punctate spots). ROIs were centered on the brightest green pixel in the green channel and average intensity was measured for red and green channels. Average background fluorescence was determined from several larger ROIs placed across the imaging field where there were no transfected axons. The average green and red background fluorescence was subtracted from the respective values before calculating R/G ratios or ΔF/F.

**Electrophysiology in slice cultures**. Hippocampal slice cultures were placed in the recording chamber of the two-photon laser scanning microscope and continuously perfused with an artificial cerebrospinal fluid (ACSF) saturated with 95% $O_2$ and 5% $CO_2$ consisting of (in mM): 119 NaCl, 26.2 NaHCO$_3$, 11 D-glucose, 1 NaH$_2$PO$_4$, 2.5 KCl, 4 CaCl$_2$, 4 MgCl$_2$ (pH 7.4, 308 mOsm) at room temperature (21-23 °C) or with a HEPES-buffered solution (in mM): 135 NaCl, 2.5 KCl, 10 Na-HEPES, 12.5 D-glucose, 1.25 NaH$_2$PO$_4$, 4 CaCl$_2$, 4 MgCl$_2$ (pH 7.4). Whole-cell recordings from CA1 pyramidal neurons were made with patch pipettes (3–4 MΩ) filled with (in mM): 135 K-gluconate, 4 MgCl$_2$, 4 Na$_2$-ATP, 0.4 Na-GTP, 10 Na$_2$-phosphocreatine, 3 sodium-L-ascorbate, and 10 HEPES (pH 7.2, 295 mOsm kg$^{-1}$). In Supplementary Fig. 4f, patch pipettes contained (in mM): 135 Cs-MeSO$_3$, 4 MgCl$_2$, 4 Na$_2$-ATP, 0.4 Na-GTP, 10 Na$_2$-phosphocreatine, 3 sodium-L-ascorbate, 10 HEPES (pH 7.2, 295 mOsm kg$^{-1}$). Series resistance was below 20 MΩ. A Multiclamp 700B amplifier (Molecular Devices) was used under the control of Ephus[58] or Wavesurfer software written in Matlab (The MathWorks). When using somatic current injection to evoke action potentials in CaMPARI2_F391W_L398V or SynTagMA expressing neurons, the antagonists CPPene (10 μM) and NBQX (10 μM) were added to the extracellular recording solution to block synaptic transmission.

**Single cell electroporation**. At DIV 13-17, CA1 neurons in rat organotypic hippocampal slice culture were transfected by single-cell electroporation[59]. Thin-walled pipettes (~10 MΩ) were filled with intracellular K-gluconate based solution into which SynTagMA variants or CaMPARI2_ F391W_L398V, or jGCaMP7b plasmid DNA was diluted to 20 ng μl$^{-1}$. In some experiments, a plasmid encoding mCerulean was also included in the pipette at 20 ng μl$^{-1}$. All experiments were conducted 3-6 days after electroporation. For blind analysis of neurons with and without SynTagMA, the electroporation mixes were coded by a second lab member and only after all recordings and analysis were completed was the investigator unblinded. Pipettes were positioned against neurons and DNA was ejected using an Axoporator 800 A (Molecular Devices) with 50 hyperpolarizing pulses (−12 V, 0.5 ms) at 50 Hz.

**Two-photon microscopy in hippocampal slices**. Two-photon imaging was performed in rat organotypic hippocampal slice cultures and mouse acute hippocampal slices. The custom-built two-photon imaging setup was based on an Olympus BX51WI microscope equipped with LUMPlan W-IR2 60× 0.9 NA (Olympus), W Plan-Apochromat 40× 1.0 NA (Zeiss) or IRAPO 25×1.0 NA (Leica) objectives controlled by the open-source software package ScanImage[60]. Two pulsed Ti:Sapphire lasers (MaiTai DeepSee, Spectra Physics) controlled by electro-optic modulators (350-80, Conoptics) were used to excite SynTagMA green (980 nm) and red species (1040 nm), respectively. When Z-stacks of SynTagMA-expressing neurons were acquired, each plane was scanned twice, using 980 nm and 1040 nm excitation, respectively. For quantification of pre- or postsynaptic targeting, axons of CA3 cells or oblique dendrites of CA1 neurons expressing mCerulean and SynTagMA were imaged at 840 nm and 980 nm. For three-color experiments, separate stacks were taken to image mCerulean (840 nm) and SynTagMA (980 nm & 1040 nm). Emitted photons were collected through the objective and oil-immersion condenser (1.4 NA, Olympus) with two pairs of photomultiplier tubes (H7422P-40, Hamamatsu). 560 DXCR dichroic mirrors and 525/50 and 607/70 emission filters (Chroma) were used to separate green and red fluorescence. Excitation light was blocked by short-pass filters (ET700SP-2P, Chroma). Two brief violet light pulses (395 nm, 100 ms, 16 mW mm$^{-2}$, 0.1 Hz) were delivered through the objective using a Spectra X Light Engine (Lumencor) just before imaging to photoswitch the CaMPARI moiety into its bright state[10].

**Viral transduction of hippocampal slices**. Organotypic hippocampal slices were microinjected at DIV 7-11 with a viral vector containing preSynTagMA under the

control of the synapsin promoter (AAV2/9-syn-preSynTagMA), or AAV2/9 synapsin-ChR2(ET/TC)−2A-synaptophysin-mCerulean. Both viruses were prepared at the UKE vector facility. Briefly, working in a laminar air flow hood, a glass pipette was backfilled with 1 μl of the viral vector and the tip inserted in the hippocampal CA3 area. A picospritzer (Science Products) coupled to the pipette was used to deliver 3–4 short (50 ms) low pressure puffs of viral vector into the tissue. Injected slices were taken back to the incubator and imaged in the multi-photon microscope 3–4 weeks later.

**Photoconversion in hippocampal slices**. Photoconversion was achieved by delivery of violet light (395 nm, 16 mW mm$^{-2}$, duration 100 ms–2 s as indicated in figure legends) using a Spectra X Light Engine (Lumencor) coupled with a liquid light guide to the epifluorescence port of the two-photon microscope. During the violet light pulses, shutters (Uniblitz) protected the photomultiplier tubes. We typically used 100–500 ms violet light pulses repeated 15–50 times with a 1 s delay from stimulus onset to photoconvert SynTagMA (see figure legends for protocol details). Image stacks were acquired prior to and following photoconversion.

**Recording and analysis of EPSCs and cellular parameters**. CA1 pyramidal neurons expressing mCerulean alone or mCerulean plus SynTagMA were patched using only the mCerulean fluorescence to identify them. TTX 1 μM, CPPene 1–10 μM, and picrotoxin 50 μM were added to oxygenated ACSF (see above). For miniature EPSC (mEPSC) measurements, recording electrodes (3–4 MΩ) contained Cs-gluconate intracellular solution (see above). After the slice rested at least 15 min in the bath, cells were patched and held at −70 mV in the whole-cell voltage clamp configuration (no liquid junction potential correction). EPSCs were recorded 5–15 min after break-in. The Event Detection feature of Clampfit 10 (Molecular Devices) was used to detect and measure mEPSC amplitudes and inter-event intervals. For cell parameter measurements, recording electrodes contained K-gluconate solution (see above). Membrane resistance ($R_m$) and capacitance ($C_m$) was calculated by Ephus using −5 mV voltage steps (50 ms) from a holding potential of −70 mV. Resting membrane voltage was measured in current clamp mode. A 1 s current ramp (0 to +600 pA) was injected to measure the rheobase and the voltage threshold for action potentials. Firing rates were calculated from 1 s current steps (−400 pA to +600 pA).

**Spine density measurement and analysis**. Two-photon microscopy at 840 nm was used to excite cells expressing mCerulean or mCerulean plus SyntagMA. An Olympus LUMFL N 60×1.1NA objective (PSF: 0.35 × 0.35 × 1.5 μm) was used to collect Z-stacks (0.3 μm z-step) of *stratum radiatum* proximal oblique dendrites (~100 μm from soma). Image stacks were deconvolved using a blind deconvolution algorithm in Autoquant X3 (Media cybernetics). Spines and dendrites were semi-automatically detected using the Filament tracer feature of Imaris (Oxford Instruments). For each neuron, spine density was determined from 1–3 dendrites of 30–100 μm length (Supplementary Fig. 4g).

**Quantification of pre- and postSynTagMA localization**. A macro written in Fiji[21] was used for two-photon 3D image analysis at 840, 980 and/or 1040 nm wavelengths. When z-stacks contained alternating images collected at different excitation wavelengths, they were first separated and then xyz-aligned to correct for chromatic aberration using green and/or red channels and the pairwise stitching plugin[61]. mCerulean and SynTagMA fluorescence values were obtained from images after median filtering and rolling ball background subtraction[62]. Regions of interest (ROI) were drawn onto maximum intensity projections and compared to axonal or dendrite shafts, respectively. Only spines projecting laterally from the dendrite were analyzed.

**Photoconversion after back-propagating action potentials**. The first image stack was acquired before patching the SynTagMA-expressing neuron (Figs. 4–5). The cell was then whole-cell patch-clamped and bAPs (100 Hz) were evoked by somatic current injection and paired with 500 ms of 395 nm light (1 s delay) in the presence of CPPene (10 μM) and NBQX (10 μM) to block synaptic transmission. This pairing was repeated 15 times. For experiments with CaMPARI2 (Fig. 2e), 50 bAPs at 100 Hz were evoked and paired with 100 ms of 395 nm light. This pairing was repeated 10 times. While acquiring the second image stack, the cell was held in voltage clamp at −65 mV to prevent depolarization. The field of violet light illumination for these experiments was 557 um in diameter and therefore large enough to illuminate the entire dendritic arbor.

**Propidium iodide staining**. Organotypic hippocampal slices aged between DIV 18-28 were used. 2-4 days before staining, CA1 neurons were electroporated with mCerulean (20 ng μl$^{-1}$) and postSynTagMA (20 ng μl$^{-1}$). Slices were placed in the two-photon imaging chamber in HEPES-buffered solution containing 3 μM propidium iodide (PI). The objective was then placed above the CA1 region of the slice and illuminated with violet light (395 nm, 16 mW mm$^{-2}$). Following illumination, slices were kept in the PI solution for 30-45 minutes and subsequently imaged (840 nm excitation). PI fluorescence was collected through 607/70 filters and mCerulean fluorescence was collected through 525/50 filters. NMDA excitotoxicity was used as a

positive control. Slices were exposed to 1 mM NMDA in HEPES-buffered solution for 1.5 h and then labeled with PI for 30–45 min before imaging as above. To quantify violet light toxicity, PI-positive nuclei were counted in each 2P image stack (background subtracted, median filtered) and normalized to the tissue volume.

**Semi-automatic segmentation and analysis of SynTagMA.** For the initial characterization of postSynTagMA (Figs. 1–3), quantification of green to red conversion was performed manually in a small set of synapses (<50) in Matlab and/or Fiji. Covering relative larger areas of the dendritic tree gave rise to larger data sets (>1000 synapses) that made a semi-automated analysis approach necessary (Figs. 4–6). Rigid registration was performed using the Pairwise stitching plugin[61] in Fiji. The broad emission spectrum of autofluorescent objects proved useful to align the red channel from the 980 nm stack to the green channel from the 1040 nm stack. Image stacks were further processed using a blind deconvolution algorithm in AutoQuant X3. Particularly critical was to correct for position change of synapses in 3D due to tissue deformation ("warping") between acquisitions of image stacks (pre- and post-stimulation). To correct for tissue warping, synapse voxels were reassigned to their initial position using either elastix[23] or ANTs (Advanced Normalization Tools[63]). To detect synapses, we used Imaris 3D spot detection on the deconvolved $G_0$ signal. Prior to data extraction, spots that were not due to SynTagMA but autofluorescence were filtered out based on their red fluorescence at baseline ($R_0$, see Supplementary Fig. 8). Approximately 10-30% of originally detected spots were rejected using this filter. Spots (volumetric ROIs) were used to extract fluorescence intensities from green and red channels (maximum pixel values from non-deconvolved, median-filtered images) at every time point. To quantify photoconversion, green ($G_0$ and $G_1$) and red ($R_0$ and $R_1$) values were normalized to the population mean of $G_0$ and $R_0$, respectively.

**Automated segmentation and analysis of SynTagMA.** To analyze thousands of synapses in large 3D datasets, we developed a pipeline to automatically identify spots in 3D and define volumes (ROIs) for accurate measurements of fluorescence changes (Supplementary Fig. 3, Supplementary Fig. 9). After two microscope-specific pre-processing steps (de-interleaving of 980/1040 nm image stacks to separate green and red channels, rigid registration to correct for chromatic aberration and beam misalignment), the data is processed by SynapseLocator, a GUI-controlled software package written in Matlab, which calls specific subroutines (Fiji, elastix, Weka) to perform the analysis:

1. Deconvolution and transformation of post-image to pre-image using elastix
2. Random forest model, built with machine learning
3. Spot identification and ROI creation
4. User interactive visualization of fluorescent values and calculations of SynTagMA photoconversion in ROIs

The detailed workflow in SynapseLocator consists of an initial deconvolution step (Fiji, DeconvolutionLab2 plugin[22]) in which diffraction-induced blurring of the images was reduced. We used elastix[23] to register image stacks across time points (using only the green channel; $G_0$ and $G_1$). Registration proceeds in four steps (rigid transformation, rotation transformation, affine transformation, and non-rigid registration) for which we provide optimized parameter sets. The transformation was then applied to both red and green channel data (deconvolved and raw). Synapse detection involved a machine learning process accessed via Fiji (Weka, Trainable Weka Segmentation plugin[24]). Two classes are manually (by the user) identified as "spot" and "no spot" to train a random forest model considering a set of scale-invariant image features (Hessian and Laplacian, each calculated at three scales, Fiji, FeatureJ plugin). Typically, the model was robust enough to allow for analysis of all experiments from a series. A synapse is identified as a region in which a group of voxels with spot class properties shows minimal connectivity (26-connected neighborhood). For each region, an ellipsoid enclosing the identified connected voxels is calculated in Matlab and stored as ROI. To exclude spots that are not related to SynTagMA fluorescence, all ROIs with high red fluorescence at baseline are automatically excluded from analysis. After spot identification, the user can interactively examine the data using a table and apply additional filters. Spot intensities are extracted in parallel from raw imaging data, median-filtered data, and from filtered and deconvolved data, allowing the user to compare the effects of image processing. Please see Github documentation for further details.

**Distance-dependence of bAP-induced calcium transients.** CA1 neurons expressing mCerulean and postSynTagMA were used for these experiments. Prior to patching, z-stacks (as described above) were acquired along the apical dendritic tree to image postSynTagMA. Two to three stacks were required to image the apical dendritic tree. The time taken to image all stacks was 30–40 min. The cell was then whole-cell patch-clamped and 50 bAPs were paired with 500 ms violet light (1 s delay). This pairing was repeated 15 times (0.1 Hz). Immediately after, all image stacks were acquired followed by complementary stacks of mCerulean at 840 nm for morphology.

In CA1 neurons expressing jGCaMP7b, frame scans ($6 \times 6$ µm) of oblique dendrites along the entire dendritic arbor were acquired (980 nm) while evoking 50 bAPs. In each trial, 50 frames ($64 \times 64$ pixels) were acquired at 5.9 Hz. At least 3 trials were recorded from each section of dendrite. Following calcium imaging, the

morphology of the entire cell was imaged using jGCaMP7b baseline fluorescence. The fluorescence time course was measured by placing ROIs on individual spines in Fiji. We calculated $\Delta F/F_0$; $F_0$ was determined 300 ms prior to bAP onset.

To analyze photoconversion (postSynTagMA) or calcium transients (jGCaMP7b) in synapses as a function of distance from the soma, we created a series of 20 µm wide concentric rings around the soma using a custom-written MATLAB script. For the synapses located in each ring, we calculated the median photoconversion or median calcium transient amplitude and the interquartile range.

**postSynTagMA turnover measurements.** postSynTagMA turnover experiments were performed on dissociated hippocampal neurons (DIV 14-16, 9-11 days post-transfection). Cells were imaged at ~21°C in a modified Tyrode's solution containing the following (in mM): 119 NaCl, 2.5 KCl, 3 CaCl2, 1 MgCl2, 25 HEPES, and 30 D-Glucose with 10 µM CNQX. PostSynTagMA photoconversion was induced using a train of 50 action potentials (50 Hz) paired with a 500 ms photoconverting light pulse in custom-drawn rectangular regions (18.6 mW mm$^{-2}$, 500-750 ms delay). Z-stacks with 0.25 µm step size were collected over the course of an hour every 5–6 min. Cells were illuminated at each step by 405 nm (1% intensity), 488 (25% intensity) and 561 nm (20% intensity) lasers.

**Photoconverting sub- and suprathreshold responses.** For extracellular synaptic stimulation, a monopolar electrode was placed in *stratum radiatum* and two 0.2 ms pulses, 40 ms apart, were delivered using an ISO-Flex stimulator (A.M.P.I.). Stimulation intensity was adjusted to be subthreshold for action potentials (i.e., to evoke ~15 mV EPSPs or ~−500 pA EPSCs) or suprathreshold (i.e. evoking action potentials) by patching a nearby neuron in CA1. Stimulation was paired with 100 ms violet light (1 s delay) 50 times.

**Subthreshold optogenetic stimulation and photoconversion.** CA3 neurons expressing AAV2/9 synapsin-ChR2(ET/TC)−2A-synaptophysin-mCerulean were stimulated with blue light pulses (470 nm, 2 pulses, 2-5 ms duration, 40 ms apart) applied through a light fiber placed above the CA3. The light intensity and pulse width were set to evoke EPSCs (0.5–1.0 nA) in a neighboring CA1 neuron. Stimulation was paired with 100 ms violet light (1 s delay) and repeated 50 times at 0.1 Hz to photoconvert postSynTagMA in active spines. Z-stacks were acquired before and after photoconversion as described above. A third z-stack at 840 nm was acquired immediately after the post-photoconversion stack to image mCerulean-labeled boutons.

**Relabeling postSynTagMA.** For relabeling experiments, paired-pulse extracellular stimulation (0.2 ms pulses, 40 ms apart) was combined with 250 ms of violet light. This was repeated 25 times (0.1 Hz) and a second image stack was acquired immediately thereafter. After imaging, the slice was returned to the incubator. Approximately 18 h later, the procedure was repeated.

**Virus injection and hippocampal window surgery.** Mice were anesthetized with an intraperitoneal injection of Ketamine/Xylazine (0.13/0.01 mg g$^{-1}$ bodyweight) and placed on a heating blanket to maintain the body temperature. In addition, mice received a subcutaneous dose of Carprofen (4 mg kg$^{-1}$) for post-surgery analgesia. Eyes were covered with eye ointment (Vidisic, Bausch + Lomb) to prevent drying. Prior to surgery, the depth of anesthesia and analgesia was evaluated with a toe-pinch to test the paw-withdrawal reflex. Subsequently, mice were fixed in a stereotactic frame, the fur was removed with a fine trimmer and the skin of the head was disinfected with Betaisodona solution using sterile cotton swabs. The skin was removed by a midline scalp incision (1–3 cm), the skull was cleaned using a bone scraper (Fine Science Tools) and a small hole was drilled with a dental drill (Foredom) above the injection site. AAV2/9-mDlx-SynTagMA-2A-mCerulean or AAV2/9-syn-GCaMP6f was targeted unilaterally to the dorsal CA1 area (−2.0 mm AP, ± 1.3 mm ML, −1.5 mm DV relative to Bregma. 0.6 µl of virus suspension was injected. All injections were done at 100 nl min$^{-1}$ using a glass micropipette. After the injection, the pipette stayed in place for at least 5 min after virus delivery before it was withdrawn and the scalp was closed with sutures. During the two days following surgery animals were provided with Meloxicam mixed into soft food.

Two weeks after virus injection, mice were anesthetized as described above to implant the hippocampal window. After fur removal, skin above the frontal and parietal bones of the skull was removed by one horizontal cut along basis of skull and two rostral cuts. The skull was cleaned after removal of the periosteum, roughened with a bone scraper and covered with a thin layer of cyanoacrylate glue (Cyano Veneer). After polymerization a 3-mm circle was marked on the right parietal bone (anteroposterior, −2.2 mm; mediolateral, +1.8 mm relative to bregma) with a biopsy punch and the bone was removed with a dental drill (Foredom). The bone fragment and dura were carefully removed with fine surgical forceps. The cortex above the hippocampus was aspirated with a 0.8 mm blunt needle connected to a water jet pump. When the first layer of fibers running orthogonal to midline became visible, a 0.4 mm blunt needle was used to carefully remove the upper two of three fiber layers of the external capsule. The craniotomy was washed with sterile PBS and a custom-built imaging window was inserted over

the dorsal hippocampus. The window consisted of a hollow glass cylinder (diameter: 3 mm, wall thickness: 0.1 mm, height: 1.5 mm) glued to a No. 1 coverslip (diameter: 3 mm, thickness: 0.17 mm) on the bottom and to a stainless-steel rim on the top with UV-curable glass glue (Norland NOA61). The steel rim and a head holder plate (Luigs & Neumann) were fixed to the skull with cyanoacrylate gel (Pattex). After polymerization, cranial window and head holder plate were covered with dental cement (Super Bond C&B, Sun Medical) to provide strong bonding to the skull bone. During the two days following surgery animals were provided with Meloxicam mixed into soft food.

**Hippocampal imaging and photoconversion in vivo.** SynTagMA imaging in vivo was performed in head-fixed animals during a behavioral task or under anesthesia (isoflurane or ketamine/xylazine). During anesthesia, mice were kept on a heated blanket to maintain body temperature and eyes were covered with eye ointment (Vidisic, Bausch + Lomb) to prevent drying. The window was centered under the two-photon microscope (MOM-scope, Sutter Instruments, modified by Rapp Optoelectronics) using a low-magnification objective (4× Nikon Plan Fluorite) and reporter expression (postSynTagMA or GCaMP6f) was verified in the hippocampus using epi-fluorescence. The cranial window was then filled with water to image in two-photon mode through a 40× (40X Nikon CFI APO NIR, 0.80 NA, 3.5 mm WD) or a 16× (16X Nikon CFI LWD Plan Fluorite Objective, 0.80 NA, 3.0 mm WD) objective. The green species of SynTagMA or GCaMP6f was excited with a Ti:Sa laser (Chameleon Vision-S, Coherent) tuned to 980 nm. The red species of SynTagMA was excited with an ytterbium-doped 1070-nm fiber laser (Fidelity-2, Coherent). Single planes (512×512 pixels) were acquired at 30 Hz with a resonant-galvanometric scanner at 29–60 mW (980 nm) and 41–60 mW (1070 nm) using ScanImage 2017b (Vidrio). Emitted photons were detected by a pair of photo-multiplier tubes (H7422P-40, Hamamatsu). A 560 DXCR dichroic mirror and 525/50 and 607/70 emission filters (Chroma Technology) were used to separate green and red fluorescence. Excitation light was blocked by a short-pass filter (ET700SP-2P, Chroma).

For identification of active CA1 neurons (Fig. 7a–c), the conversion protocol consisted of 2 s violet light pulses (405 nm, 12.1 mW mm$^{-2}$) repeated 10 or 15 times in awake and ketamine/xylazine (130/10 mg/kg) anesthetized mice. In awake behaving mice, photoconversion light was triggered upon water reward delivery. Acute imaging of calcium-dependent dimming of the green SynTagMA species was performed continuously while the animals were running multiple laps on the treadmill receiving a water reward upon completion of each lap. Running speed was simultaneously recorded.

For synaptic imaging of SynTagMA in hippocampal interneurons mice were anesthetized with isoflurane at a concentration ranging between 2.0% and 2.5% in 100% O$_2$. Since SynTagMA fluorescence was relatively low, 450–1800 frames per optical plane were acquired and averaged. Under isoflurane anesthesia (Fig. 7g), conversion of active synapses in interneurons was achieved with 20 flashes of 3 s violet light pulses (405 nm, 0.42 mW mm$^{-2}$). Motion artefacts were corrected with a custom-modified Lucas-Kanade-based alignment algorithm in Matlab. Quantification of nuclear and synaptic SynTagMA photoconversion was performed as described for slice cultures. Raw movies from acute SynTagMA imaging experiments were analyzed with suite2p, which performs image registration (motion correction), automatic region-of-interest detection, activity extraction and neuropil correction.

**Quantification and statistical analysis.** Quantification was done using automatic (i.e., blind) analyses with either Imaris or SynapseLocator, or quantification was done manually. Figures 1 and 2 were quantified manually (see Supplementary Fig. 9 for comparison of manual versus automatic analysis). The n-size for Figs. 1 and 2 is neuron number (all independent replications). However, some experiments use number of synapses as the n-size. For all in vitro postSynTagMA experiments (Figs. 3–6), the n-size is synapse number. We consider every synapse a unique biological entity. In vivo experiments use nuclei or synapses as the n-size (Fig. 7). The definition and the exact value for n is given in the figure and/or figure legend. Replication of experiments is reported in figure legends. Statistical analysis was performed using GraphPad Prism (v8) or Matlab. For data with normal distributions, Student's $t$-test or one-way ANOVA followed by Tukey's post-hoc comparison were used. Data considered non-normal (according to a D'Agostino-Pearson test) underwent non-parametric tests (Kruskal-Wallis followed by Benjamini-Hochberg FDR method, Mann-Whitney test Dunn's multiple comparisons test). Curve fitting was done in GraphPad Prism or in Matlab using the curve-fitting toolbox. Data are shown as either mean ± SEM or median with interquantile range. The specific test used is reported in the corresponding figure legend.

**Reporting summary.** Further information on research design is available in the Nature Research Reporting Summary linked to this article.

## Data availability
The datasets generated and/or analyzed during the study are available from the corresponding author TGO on request. An example preSynTagMA and postSynTagMA dataset is deposited on Github with SynapseLocator.

## Code availability
We deposited our analysis software, instructions and test data on GitHub (https://github.com/drchrisch/SynapseLocator).

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

## Acknowledgements

We thank Iris Ohmert, Sabine Graf and Kathrin Sauter for excellent technical assistance. Ingke Braren of the UKE Vector Facility produced AAV vectors. mCerulean was kindly provided by Wolfgang Wagner at the ZMNH. PSD95.FingR was kindly provided by Don Arnold at USC. Synaptophysin-GCaMP3 was kindly provided by Loren Looger at HHMI's Janelia Farm. Funding was received from: The Deutsche Forschungsgemeinschaft (DFG) grants SPP 1665 220176618, SPP1926 273915538, SFB 936 178316478, SFB 1328 335447717, FOR 2419 278170285; the European Research Council (ERC-2016-StG 714762); the National Institute of Health (GM13132); the Howard Hughes Medical Institute. BCF was supported by an individual fellowship from the DAAD (Research Grants - Doctoral Programmes in Germany, 57214224). RJO was supported by the Esther and Joseph Klingenstein-Simons Foundation (FP00003669) and U.S. Department of Education GAANN grant (P200A150059). LCP was supported by NSF IOS grant 1750199. BM holds a postdoctoral fellowship from the Research Foundation-Flanders (FWO Vlaanderen). WY was supported by the China Scholarship Council (CSC 201606210129).

## Author contributions

Conceptualization, T.G.O., C.E.G., M.B.H. and J.S.W.; Software, A.P.A., C.S. and I.A.C.; Investigation, A.P.A., B.C.F., R.J.O., W.Y., P.L.M., B.M., M.A.M., L.C.P., J.S.W. and M.B.H.; Resources, E.R.S.; Writing – Original Draft, C.E.G., T.G.O., M.B.H., J.S.W., B.C.F., A.P.A., C.S.; Writing – Review & Editing, A.P.A., B.C.F., R.J.O., W.Y., I.A.C., P.L.M., B.M., M.A.M., L.C.P., C.S., E.R.S., J.S.W., C.E.G., M.B.H., T.G.O.; Visualization, B.C.F., A.P.A., C.S., T.G.O., M.B.H.; Supervision, E.R.S., J.S.W., C.E.G., M.B.H., T.G.O.; Project administration, C.E.G.; Funding acquisition, E.R.S., J.S.W., C.E.G., M.B.H., T.G.O.

## Competing interests

E.R.S. is an inventor on US patent number 9,518,996 and US patent application 15/335,707, which may cover CaMPARI sequences described in this paper. The remaining authors declare no competing interests.

## Additional information

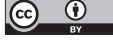

