## [Peer Review File · Nature Communications]

Reviewers' comments:

Reviewer #1 (Remarks to the Author):

In their manuscript, “Freeze-frame imaging of synaptic activity using SynTagMA”, Perez-Alvarez et al. develop and test a strategy for fluorescently tagging active synapses for subsequent spatial analysis by optical microscopy. For tagging presynaptic terminals, the authors conjugate a photoactivatable, calcium-sensitive fluorescent protein variant (CaMPARI2) to presynaptic synaptophysin and use this probe to transfect cultured hippocampal cells in vitro and CA1 hippocampal neurons in situ. For labeling postsynaptic terminals, the authors expressed constructs encoding CaMPARI2-intrabody fusion proteins that bind to PSD95 and tested these probes in vitro, in situ, and in vivo. To identify active synapses, the authors use ultraviolet light stimulation to convert Ca²⁺-bound CaMPARI2 from a green to a red fluorescent state. The authors present data from experiments conducted in vitro, in situ, and in vivo demonstrating photoconversion of postSynTagMA in dendritic spines and cell nuclei in response to several stimulation protocols (backpropagating APs, ChR2+ afferent stimulation, behavior). Quantitative analysis and raw imaging results showing the kinetics of preSynTagMA photoconversion in vitro are shown. The authors developed automated image analysis pipelines to quantify active synapses and benchmarked these approaches against a subsample of synapses annotated manually. The data are quantitative, experimental results from several methods of analysis (in vitro, in situ, and in vivo) are presented, and proper attention is given to strengths/limitations of the approach. To facilitate replication of the experiments, the reagents used/developed are clearly described and the analysis code is made available electronically. The problem of marking active synapses in neural circuits is a grand challenge, and this paper describes a set of molecular/genetic/analysis tools that could be used to provide snapshots (images) and automated detection/analysis of active synapses over large tissue volumes. The results of this work will be of interest to researchers who are using Ca²⁺ and voltage indicators to map neural activity patterns and for a wider field of physiologists investigating neural circuits using transgenic-labeling approaches. In this reviewer’s opinion, with minor revisions to improve the clarity of the text/figures/analysis (see below) this manuscript will be suitable for publication at Nature Communications.

Minor suggestions for revision:

- 1) References to “V light” in the text and figures should be changed to “UV light” for clarity
- 2) In Figure 1E, the purple colors and line style of the Control versus TTX conditions are very similar and should be changed to either different colors, different line style (e.g. dashed versus solid), or some combination of both for clarity
- 3) In the introduction, the authors write “1) We improved the brightness and conversion efficiency of CaMPARI”. For clarity, I suggest expanding this sentence to introduce the name of the new molecule and the modification used to generate it. E.G. “We introduced point mutations in CaMPARI to generate a new probe, CaMPARI2, with improved brightness and conversion efficiency”
- 4) The presentation and organization of Supplementary Figure 1 is confusing. In the body of the manuscript, the authors reference figure panels out of order (Supplementary Figure 1H is referenced

before any other panels in this figure). Further, Supplementary Figure 1H references Supplemental Figure 1D (within the legend) and the reader must read and interpret the entire set of figures panels/experiments without clear logical direction from the manuscript. Additionally, supplementary Figure 1 Panels A/B/C are never referenced in the text. To correct these issues, the authors should reorganize either the text or the figure layout, or both, to have all figures and panels in the manuscript introduced sequentially. All experiments appearing in supplementary data should be clearly (albeit briefly) described, discussed, and referenced in the main results section of the manuscript.

5) The authors write “The important stabilizing function of autoregulation was confirmed in viral expression experiments (Supplementary Fig. 2).” The phrase “stabilizing function of autoregulation” is confusing and should be changed to jargon-free language for clarity.

6) The authors write “We evoked backpropagating action potentials (bAPs) by brief somatic current injections to raise intracellular Ca²⁺, and applied 395 nm light pulses (500 ms) with a 1 s delay to characterize postSynTagMA photoconversion.” The choice of a 1s delay appears to be based on the kinetics measured for preSynTagMA (Figure 2E). Do the authors know that the kinetics of CaMPARI2 photoconversion (temporal window) is similar in presynaptic versus postsynaptic compartments? And, to what extent does this depend on the Ca²⁺ affinity of the selected CaMPARI2 variant given that Ca²⁺ concentration/kinetics are expected to be different in pre versus post synaptic compartments? If possible these points should be addressed, ideally with data showing the kinetics of postSynTagMA photoconversion. At the least, these issues should be discussed in the manuscript. This is also relevant for the authors’ discussion of the proximity-dependent photoconversion of postSynTagMA during bAPs.

7) In the data presented, particularly from experiments *in situ* and *in vivo*, what fraction of the total postSynTagMA signal was excluded from analysis due to the presence of endogenous artefactual red fluorescence? If possible, the authors should quantify and report this exclusion fraction and update their “Limitations of the method” discussion to refer to false-negative error rates in the detection of activated synapses. The absolute density (summed area of clusters/total imaged area) of endogenous artifact (in the tissue examined) should also be investigated as it is relevant for future applications of the method that may seek to achieve dense synapse labeling and analysis.

8) There are typos in the text that should be corrected (e.g. “autofluorescent objects tuned out to be very useful” and “the data is processed”)

9) Does preSynTagMA work *in situ*? *In vivo*? Figure 2H shows localization to axonal boutons, but no data are provided demonstrating effective photoconversion at presynaptic terminals *in situ*.

Reviewer #2 (Remarks to the Author):

Perez-Alvarez et al. present two new versions of Campari2, one targeted to pre- and the other to the post-synaptic sites. Presynaptic targeting was engineered by fusing Campari to synaptophysin. Presynaptic Campari had many of the same features as untargeted Campari in that it dimmed with repeated action potentials and was capable of photoconversion with simultaneous Ca influx and violet light exposure. Also, it was maximally photoconverted when the violet pulse followed tetanic stimulation by 2-5s. To improve the temporal properties and reduce photoconversion due to exposure

to violet light or high Ca²⁺ alone, Campari was replaced with Campari2 to give SynTagMA. SynTagMA was maximally activated 200 ms following tetanic stimulation and was not activated by exposure to Ca²⁺ or to violet light alone. In order to target Campari2 postsynaptically, it was fused to PSD95.FingR, an intrabody that binds to the synaptic scaffolding protein PSD95, and combined with a slightly modified version of a transcriptional regulator developed in conjunction with PSD95.FingR that virtually eliminates unbound FingR. Control experiments verified that expression of the resulting construct, postSynTagMA, did not affect spine morphology or synaptic transmission. They also showed that postSynTagMA expressed in cultured hippocampal slices underwent photoconversion in response to backpropagating action potentials and that using the sum of the green fluorescence before and after as the denominator gives a measure of photoconversion that is relatively independent of spine size. In addition, the authors developed novel software that can be used to automatically map the amounts and locations of photoconversion at individual synapses, dramatically reducing the amount of time required to analyzed data vs. manual analysis. They showed that photoconversion of postSynTagMA in response to backpropagating APs decreased exponentially with distance from the soma, which correlates to the drop off in Ca²⁺ signals measured with GCaMP. In slice cultures they showed that a postSynTagMA could be photoconverted in response to presynaptic stimulation, that photoconversion decayed with a time constant of 30 minutes, and that a synapse could be re-photoconverted. Finally, they showed that postSynTagMA could be photoconverted at active synapses in vivo in an actively behaving mouse.

This paper describes two new modifications of Campari that could overcome some of the limitations of the original, and have the potential to be extremely useful, particularly when combined with the innovative new software platform that is described in the paper. On the whole the constructs appear to have been tested in a thorough and careful manner. I have just a few queries/requests that I have highlighted below.

1. Do you see enlargement of puncta labeled with PreSynTagMA over time? What is the maximum amount of time the PreSynTagMA can be expressed before toxicity is seen?
2. Could you provide more validation of the program? What percentage of synapses were identified by both the program and by human analysts? Why was the slope of the graph in Supplementary Fig. 6b not equal to 1? Does the deconvolution preserve the intensities of puncta?
3. What is the maximum depth at which violet light can photoconvert SynTagMA in vivo? Presumably the scattering of violet light will be a problem with photoconversion deep within tissue. Could this be mitigated by using fiber optic light delivery as is done to activate Channelrhodopsin?
4. How can you know that a specific area is inactive, vs. whether it has not received sufficient violet light for photoconversion?

Reviewers' comments:

Reviewer #1 (Remarks to the Author):

In their manuscript, “Freeze-frame imaging of synaptic activity using SynTagMA”, Perez-Alvarez et al. develop and test a strategy for fluorescently tagging active synapses for subsequent spatial analysis by optical microscopy. For tagging presynaptic terminals, the authors conjugate a photoactivatable, calcium-sensitive fluorescent protein variant (CaMPARI2) to presynaptic synaptophysin and use this probe to transfect cultured hippocampal cells in vitro and CA1 hippocampal neurons in situ. For labeling postsynaptic terminals, the authors expressed constructs encoding CaMPARI2-intrabody fusion proteins that bind to PSD95 and tested these probes in vitro, in situ, and in vivo. To identify active synapses, the authors use ultraviolet light stimulation to convert Ca²⁺-bound CaMPARI2 from a green to a red fluorescent state. The authors present data from experiments conducted in vitro, in situ, and in vivo demonstrating photoconversion of postSynTagMA in dendritic spines and cell nuclei in response to several stimulation protocols (backpropagating APs, ChR2+ afferent stimulation, behavior). Quantitative analysis and raw imaging results showing the kinetics of preSynTagMA photoconversion in vitro are shown. The authors developed automated image analysis pipelines to quantify active synapses and benchmarked these approaches against a subsample of synapses annotated manually. The data are quantitative, experimental results from several methods of analysis (in vitro, in situ, and in vivo) are presented, and proper attention is given to strengths/limitations of the approach. To facilitate replication of the experiments, the reagents used/developed are clearly described and the analysis code is made available electronically. The problem of marking active synapses in neural circuits is a grand challenge, and this paper describes a set of molecular/genetic/analysis tools that could be used to provide snapshots (images) and automated detection/analysis of active synapses over large tissue volumes. The results of this work will be of interest to researchers who are using Ca²⁺ and voltage indicators to map neural activity patterns and for a wider field of physiologists investigating neural circuits using transgenic-labeling approaches. In this reviewer’s opinion, with minor revisions to improve the clarity of the text/figures/analysis (see below) this manuscript will be suitable for publication at Nature Communications.

Minor suggestions for revision:

1) References to “V light” in the text and figures should be changed to “UV light” for clarity

405 nm light is well inside the violet range (380-450 nm) and outside UV-A (315-380). For clarity, we replaced the non-standard abbreviation ‘V light’ with ‘violet light’ or ‘violet’ in the text and in all figures.

2) In Figure 1E, the purple colors and line style of the Control versus TTX conditions are very similar and should be changed to either different colors, different line style (e.g. dashed versus solid), or some combination of both for clarity

We changed the plot accordingly, TTX condition lines and markers are now red.

3) In the introduction, the authors write “1) We improved the brightness and conversion efficiency of CaMPARI”. For clarity, I suggest expanding this sentence to introduce the name of the new molecule and the modification used to generate it. E.G. “We introduced point mutations in CaMPARI to generate a new probe, CaMPARI2, with improved brightness and conversion efficiency”

We expanded the sentence according to the suggestion (page 2, line 50).

4) The presentation and organization of Supplementary Figure 1 is confusing. In the body of the manuscript, the authors reference figure panels out of order (Supplementary Figure 1H is referenced before any other panels in this figure). Further, Supplementary Figure 1H references Supplementary Figure 1D (within the legend) and the reader must read and interpret the entire set of figures panels/experiments without clear logical direction from the manuscript. Additionally, supplementary Figure 1 Panels A/B/C are never referenced in the text. To correct these issues, the authors should reorganize either the text or the figure layout, or both, to have all figures and panels in the manuscript introduced sequentially. All experiments appearing in supplementary data should be clearly (albeit briefly) described, discussed, and referenced in the main results section of the manuscript.

We now cite all Supplementary figure panels in sequence (page 4, lines 99-107):

PreSynTagMA showed no photoconversion in the absence of activity (Supplementary Fig. 1a-c). We could readily distinguish active from inactive axons by the differential preSynTagMA photoconversion in the presence of the GABA_A antagonist bicuculline (Supplementary Fig. 1d-g). When the same axons were directly electrically stimulated via field electrodes, green preSynTagMA fluorescence dimmed in all axons, indicating that action potentials triggered calcium transients in all axons (Supplementary Fig. 1h). The rapid recovery from dimming corresponds to the short photoconversion time window (0.2 – 2 s post-stimulation) of preSynTagMA (Supplementary Fig. 1h, Fig. 2d, e).

5) The authors write “The important stabilizing function of autoregulation was confirmed in viral expression experiments (Supplementary Fig. 2).” The phrase “stabilizing function of autoregulation” is confusing and should be changed to jargon-free language for clarity.

We expanded the section and reordered the supplementary figures to clarify. Supplementary Fig. 5 (formerly Supp. Fig. 2) now better shows and explains the goal of viral expression (page 5, lines 139-143):

For global labeling experiments, we generated a recombinant AAV9 encoding postSynTagMA, which produced dense punctate expression throughout the neuropil (Supplementary Fig. 5a-d). Viral expression of the construct without autoregulation (no ZF-KRAB(A)) flooded the neurons with fluorescent protein and was not usable for synaptic imaging (Supplementary Fig. 5e-g).

6) The authors write “We evoked backpropagating action potentials (bAPs) by brief somatic current injections to raise intracellular Ca²⁺, and applied 395 nm light pulses (500 ms) with a 1 s delay to characterize postSynTagMA photoconversion.” The choice of a 1s delay appears to be based on the kinetics measured for preSynTagMA (Figure 2E). Do the authors know that the kinetics of CaMPARI2 photoconversion (temporal window) is similar in presynaptic versus postsynaptic compartments? And, to what extent does this depend on the Ca²⁺ affinity of the selected CaMPARI2 variant given that Ca²⁺

concentration/kinetics are expected to be different in pre versus post synaptic compartments? If possible these points should be addressed, ideally with data showing the kinetics of postSynTagMA photoconversion. At the least, these issues should be discussed in the manuscript. This is also relevant for the authors' discussion of the proximity-dependent photoconversion of postSynTagMA during bAPs.

In response to this query, we measured the optimal light pulse delay for CaMPARI2 (F391W/F398V) in dendrites, the same variant used in both pre- and post-SynTagMA. The results are plotted in Fig. 2e overlaid (white-fill triangles) with the preSynTagMA data (see text: page 4 lines 105-107). The photoconversion time window of (soluble) CaMPARI2(F391W/F398V) in dendrites was near-identical to preSynTagMA and to our published characterization of (lower affinity) CaMPARI2 photoconversion (Moeyaert et al., 2018). Therefore, we don't think targeting has significant effects on the kinetics of photoconversion.

7) In the data presented, particularly from experiments in situ and in vivo, what fraction of the total postSynTagMA signal was excluded from analysis due to the presence of endogenous artefactual red fluorescence? If possible, the authors should quantify and report this exclusion fraction and update their "Limitations of the method" discussion to refer to false-negative error rates in the detection of activated synapses. The absolute density (summed area of clusters/total imaged area) of endogenous artifact (in the tissue examined) should also be investigated as it is relevant for future applications of the method that may seek to achieve dense synapse labeling and analysis.

In response, we added the new **Supplementary Figure 8** to better explain the effects of autofluorescence on spot detection and analysis. As we state in the methods, we rejected 10-30% of detected spots based on their high red fluorescence at baseline. Almost all of these spots are not related to SynTagMA fluorescence, but were erroneously detected in highly autofluorescent areas far from any SynTagMA-labeled neuron (**Supplementary Figure 8c**). In rare cases, a SynTagMA-labeled spine can be very close to autofluorescence, leading to elevated R_0 values and subsequent rejection (**Supplementary Figure 8d, e**).

To estimate the probability of such non-analyzable synapses, we calculated the volume content of red autofluorescence using the surface feature in Imaris and compared its volume to the total imaged volume (Reviewer Fig. 1). We added the following section to the discussion (p. 11 line 324-330):

SynTagMA analysis is complicated by endogenous red fluorescence which can be mistaken for photoconverted SynTagMA if only a single time point is considered. We found that depending on the age of the tissue, 0.4-4% of the total imaged volume was red fluorescent at baseline. Green objects detected inside these autofluorescent areas were typically non-synaptic (Supplementary Fig. 8). For automatic exclusion, SynapseLocator sets a dynamic threshold based on the histogram of the red channel at baseline (R_0 , before photoconversion) and rejects objects with elevated R_0 . This process effectively prevents false-positive errors, but requires acquisition of at least two time points (before and after photoconversion).

Reviewer Figure 1: The volume fraction of red autofluorescence is ~1%. Thus, on average, 1 out of 100 SynTagMA-labeled synapses could not be analyzed. Examples are shown in Supplementary Figure 8d.

8) There are typos in the text that should be corrected (e.g. “autofluorescent objects tuned out to be very useful” and “the data is processed”)

Done

9) Does preSynTagMA work in situ? In vivo? Figure 2H shows localization to axonal boutons, but no data are provided demonstrating effective photoconversion at presynaptic terminals in situ.

We now show additional preSynTagMA photoconversion experiments in slice cultures in a new **Supplementary Fig. 3**. Viral transfection of CA3 cells provided excellent preSynTagMA labeling of Schaffer collateral boutons and proof for activity-dependent photoconversion. We added the following section to the results (page 4, lines 112-117):

For dense axonal labeling, we microinjected a viral vector (AAV2/9-syn-preSynTagMA) in hippocampal CA3 and imaged Schaffer collateral boutons in CA1 stratum radiatum (Supplementary Fig. 3). Repeated electrical stimulation combined with violet light illumination induced photoconversion of preSynTagMA-expressing boutons while no photoconversion was induced by identical violet illumination when action potentials were suppressed by TTX.

Reviewer #2 (Remarks to the Author):

Perez-Alvarez et al. present two new versions of Campari2, one targeted to pre- and the other to the post-synaptic sites. Presynaptic targeting was engineered by fusing Campari to synaptophysin. Presynaptic Campari had many of the same features as untargeted Campari in that it dimmed with repeated action potentials and was capable of photoconversion with simultaneous Ca influx and violet light exposure. Also, it was maximally photoconverted when the violet pulse followed tetanic stimulation by 2-5s. To improve the temporal properties and reduce photoconversion due to exposure to violet light or high Ca²⁺ alone, Campari was replaced with Campari2 to give SynTagMA. SynTagMA was maximally activated 200 ms following tetanic stimulation and was not activated by exposure to Ca²⁺ or to violet light alone. In order to target Campari2 postsynaptically, it was fused to PSD95.FingR, an intrabody that binds to the synaptic scaffolding protein PSD95, and combined with a slightly modified version of a transcriptional regulator developed in conjunction with PSD95.FingR that virtually eliminates unbound FingR. Control experiments verified that expression of the resulting construct, postSynTagMA, did not

affect spine morphology or synaptic transmission. They also showed that postSynTagMA expressed in cultured hippocampal slices underwent photoconversion in response to backpropagating action potentials and that using the sum of the green fluorescence before and after as the denominator gives a measure of photoconversion that is relatively independent of spine size. In addition, the authors developed novel software that can be used to automatically map the amounts and locations of photoconversion at individual synapses, dramatically reducing the amount of time required to analyzed data vs. manual analysis. They showed that photoconversion of postSynTagMA in response to backpropagating APs decreased exponentially with distance from the soma, which correlates to the drop off in Ca²⁺ signals measured with GCaMP. In slice cultures they showed that a postSynTagMA could be photoconverted in response to presynaptic stimulation, that photoconversion decayed with a time constant of 30 minutes, and that a synapse could be re-photoconverted. Finally, they showed that postSynTagMA could be photoconverted at active synapses in vivo in an actively behaving mouse.

This paper describes two new modifications of Campari that could overcome some of the limitations of the original, and have the potential to be extremely useful, particularly when combined with the innovative new software platform that is described in the paper. On the whole the constructs appear to have been tested in a thorough and careful manner. I have just a few queries/requests that I have highlighted below.

1. Do you see enlargement of puncta labeled with PreSynTagMA over time? What is the maximum amount of time the PreSynTagMA can be expressed before toxicity is seen?

In the first days after transfection, the brightness of preSynTagMA puncta (and therefore their apparent size) increases. As the physical size of presynaptic terminals also tends to increase with age (Dyson and Jones, 1980), it is difficult to disentangle the effects of increasing expression from biological growth. Individual boutons are at the diffraction limit of the light microscope; a definite answer would require EM level analysis which is beyond the scope of this study. As our probe is ratiometric, we don't expect the absolute brightness of puncta to affect photoconversion during activity.

We did not observe any toxic effects (beading, swelling, etc.) of prolonged preSynTagMA expression. We tested 1 week of expression in dissociated neuronal culture or up to 4 weeks in virally transduced hippocampal slices (See Suppl. Fig. 4 for a detailed quantitative comparison between SynTagMA expressing neurons and non-transfected controls). To better document expression times, we added information about the length of expression to the figures. Dedicated studies have shown that synaptophysin overexpression does not alter presynaptic properties (Granseth et al., 2006). It is considered the ideal anchor to selectively target genetically encoded probes to vesicles (Matz et al., 2010; Sankaranarayanan et al., 2000).

2. Could you provide more validation of the program? What percentage of synapses were identified by both the program and by human analysts? Why was the slope of the graph in Supplementary Fig. 6b not equal to 1?

In the previous Supplementary Fig. 6 (now Supp. Fig. 8), approximately 4000 total spots were detected by SynapseLocator and of those, 1059 spots were inside squares which received violet light at different times. In Supp. 8b, we show 413 spots which were found by both human analyst and the program, corresponding to 39% of the total 1059 spots found by SynapseLocator inside the illuminated boxes. In this example, the human analyst attempted to pick the same number of spots for each illumination condition, explaining the much smaller number of hand-picked spots. Human analysis was not meant to provide a ground-truth standard for automated synapse detection. In addition, automated analysis was restricted to the brightest voxels in the center of the detected spots while the ‘manual’ analysis used more generous ROIs encompassing the entire bouton. This explains the systematically lower average fluorescence values in the manual analysis, resulting in a slope > 1. However, the correlation between both analyses is still very high, meaning that a classification into active vs. non-active synapses (via thresholding) would result in near-identical outcomes.

Does the deconvolution preserve the intensities of puncta?

Deconvolution changes the intensities of puncta (as it should!), but the amount of change strongly depends on the particular deconvolution algorithm used (Hygens vs. AutoQuant X3 vs. ImageG/Fiji). For this publication, we chose a conservative approach, using deconvolution prior to object detection, but extracting intensities from the non-deconvolved data (as stated in the methods, line 623). We now show in the extended Supplemental Fig. 6 side-by-side analysis of non-deconvolved and deconvolved data for a direct comparison. Deconvolution greatly improved the separation of stimulated and non-stimulated PSDs, but changed (increased) the absolute values. In our opinion, depending on the goal of the experiment, the user should be able to decide which analysis strategy to use. Therefore, we modified our analysis software (SynapseLocator) to analyze raw, median filtered and deconvolved data in parallel, leaving the final decision to the user. The new version has been deposited on GitHub.

In addition to the new example (Supp. Fig. 6d), we added a sentence to the methods (Page 8, lines 652 p.): *Spot intensities are extracted in parallel from raw imaging data, median-filtered data, and from filtered and deconvolved data, allowing the user to compare the effects of image processing.*

3. What is the maximum depth at which violet light can photoconvert SynTagMA in vivo? Presumably the scattering of violet light will be a problem with photoconversion deep within tissue. Could this be mitigated by using fiber optic light delivery as is done to activate Channelrhodopsin?

We don't think fiber delivery of violet light is a good strategy, as the intensity of photoconversion light will drop steeply from the injection point. We measured the light-dependence of photoconversion and found it linear, as expected for a single photon process (new Supp. Fig. 11). In addition, we modeled light scattering in brain tissue and found the cranial window strategy superior to fiber delivery (new Supp. Fig. 11). We added the following section to the discussion (Page 12, lines 340-347):

For in vivo experiments, depth-dependent attenuation of the violet photoconversion light by scattering and absorption has to be considered. We found that photoconversion efficiency was a linear function of violet light intensity (Supplementary Fig. 11). Using published modeling software, we compared cranial window illumination (405 nm) to illumination via implanted optical fiber. While the relatively

homogenous illumination through a cranial window leads to uniform light intensity (and thus, photoconversion) within a field of view (Fig. 7a, Supplementary Fig. 11), photoconversion via fiber would be expected to be rather inhomogeneous in all spatial directions and results therefore hard to interpret. Similar to CaMPARI experiments, it is safest to compare cells or synapses imaged at similar depths (Trojanowski and Turrigiano, 2019).

4. How can you know that a specific area is inactive, vs. whether it has not received sufficient violet light for photoconversion?

The problem of variability induced by uneven excitation is not specific to SynTagMA, but applies to all optical sensors (e.g. CaMPARI, GCaMP). It has not really prevented meaningful experiments with read-out of activity. In general, it is safest to compare cells or synapses imaged at the same depth. In this case, differential photoconversion cannot be due to differences in illumination. This strategy has been employed in CaMPARI experiments (Trojanowski and Turrigiano, 2019). The strong scattering of violet light in brain tissue acts as a diffusor, averaging out local inhomogeneities of the light source with increasing depth. Thus, if photoconverted synapses (or neurons) are found next to non-photoconverted ones, it is reasonable to assume that the lack of photoconversion was due to low activity. Large diameter collimated light sources are best suited to ensure homogenous photoconversion.

References

Dyson, S.E., and Jones, D.G. (1980). Quantitation of terminal parameters and their interrelationships in maturing central synapses: A perspective for experimental studies. *Brain Res.* *183*, 43–59.

Granseth, B., Odermatt, B., Royle, S.J., and Lagnado, L. (2006). Clathrin-mediated endocytosis is the dominant mechanism of vesicle retrieval at hippocampal synapses. *Neuron* *51*, 773–786.

Matz, J., Gilyan, A., Kolar, A., McCarvill, T., and Krueger, S.R. (2010). Rapid structural alterations of the active zone lead to sustained changes in neurotransmitter release. *Proc. Natl. Acad. Sci. U. S. A.* *107*, 8836–8841.

Moeyaert, B., Holt, G., Madangopal, R., Perez-Alvarez, A., Fearey, B.C., Trojanowski, N.F., Ledderose, J., Zolnik, T.A., Das, A., Patel, D., et al. (2018). Improved methods for marking active neuron populations. *Nat. Commun.* *9*, 4440.

Sankaranarayanan, S., De Angelis, D., Rothman, J.E., and Ryan, T.A. (2000). The use of pHluorins for optical measurements of presynaptic activity. *Biophys J* *79*, 2199–2208.

Trojanowski, N.F., and Turrigiano, G. (2019). Activity labeling in vivo using CaMPARI2 reveals electrophysiological differences between neurons with high and low firing rate set points. *BioRxiv* 795252.

****REVIEWERS' COMMENTS:**

Reviewer #1 (Remarks to the Author):

The authors have provided a thoughtful and careful response to each of the reviewers' questions and comments provided in the initial review.

New data have been included, which help to clarify and improve the manuscript.

New discussion points and appropriate clarifications to figures/text have been provided.

In this reviewer's opinion, the manuscript is suitable for publication in its current form.

Reviewer #2 (Remarks to the Author):

The authors have satisfied all of my concerns.